# Superspreaders drive the largest outbreaks of hospital onset COVID-19 infections

Christopher JR Illingworth[1,2,3†]*, William L Hamilton[4,5†], Ben Warne[4,5], Matthew Routledge[5,6], Ashley Popay[7], Chris Jackson[1], Tom Fieldman[4,5], Luke W Meredith[8], Charlotte J Houldcroft[4], Myra Hosmillo[8], Aminu S Jahun[8], Laura G Caller[8], Sarah L Caddy[9], Anna Yakovleva[8], Grant Hall[8], Fahad A Khokhar[4,9], Theresa Feltwell[4], Malte L Pinckert[8], Iliana Georgana[8], Yasmin Chaudhry[8], Martin D Curran[6], Surendra Parmar[6], Dominic Sparkes[5,6], Lucy Rivett[5,6], Nick K Jones[5,6], Sushmita Sridhar[4,9,10], Sally Forrest[9], Tom Dymond[5], Kayleigh Grainger[5], Chris Workman[5], Mark Ferris[5], Effrossyni Gkrania-Klotsas[5,11,12], Nicholas M Brown[5], Michael P Weekes[4,9], Stephen Baker[4,9], Sharon J Peacock[4,10,13], Ian G Goodfellow[8], Theodore Gouliouris[4,5,6], Daniela de Angelis[2,13], M Estée Török[4,5]

[1]MRC Biostatistics Unit, University of Cambridge, East Forvie Building, Forvie Site, Robinson Way, Cambridge, United Kingdom; [2]Institut für Biologische Physik, Universität zu Köln, Köln, Germany; [3]Department of Applied Mathematics and Theoretical Physics, Centre for Mathematical Sciences, Cambridge, United States; [4]University of Cambridge, Department of Medicine, Cambridge Biomedical Campus, Cambridge, United Kingdom; [5]Cambridge University Hospitals NHS Foundation Trust, Cambridge Biomedical Campus, Cambridge, United Kingdom; [6]Public Health England Clinical Microbiology and Public Health Laboratory, Cambridge Biomedical Campus, Cambridge, United Kingdom; [7]Public Health England Field Epidemiology Unit, Cambridge Institute of Public Health, Forvie Site, Cambridge Biomedical Campus, Cambridge, United Kingdom; [8]University of Cambridge, Department of Pathology, Division of Virology, Cambridge Biomedical Campus, Cambridge, United Kingdom; [9]Cambridge Institute for Therapeutic Immunology and Infectious Disease, Jeffrey Cheah Biomedical Centre, Cambridge, United Kingdom; [10]Wellcome Sanger Institute, Wellcome Trust Genome Campus, Hinxton, United Kingdom; [11]MRC Epidemiology Unit, University of Cambridge, Level 3 Institute of Metabolic Science, Cambridge, United Kingdom; [12]University of Cambridge, School of Clinical Medicine, Cambridge Biomedical Campus, Cambridge, United Kingdom; [13]Public Health England, National Infection Service, London, United Kingdom

*For correspondence: cjri2@cam.ac.uk

†These authors contributed equally to this work

Competing interests: The authors declare that no competing interests exist.

**Abstract** SARS-CoV-2 is notable both for its rapid spread, and for the heterogeneity of its patterns of transmission, with multiple published incidences of superspreading behaviour. Here, we applied a novel network reconstruction algorithm to infer patterns of viral transmission occurring between patients and health care workers (HCWs) in the largest clusters of COVID-19 infection identified during the first wave of the epidemic at Cambridge University Hospitals NHS Foundation Trust, UK. Based upon dates of individuals reporting symptoms, recorded individual locations, and viral genome sequence data, we show an uneven pattern of transmission between individuals, with patients being much more likely to be infected by other patients than by HCWs. Further, the data

were consistent with a pattern of superspreading, whereby 21% of individuals caused 80% of transmission events. Our study provides a detailed retrospective analysis of nosocomial SARS-CoV-2 transmission, and sheds light on the need for intensive and pervasive infection control procedures.

## Introduction

Reducing the spread of SARS-CoV-2 is a crucial priority for controlling and limiting the impact of the COVID-19 pandemic. Key metrics in assessing transmission are the basic and effective reproduction R numbers, which describe the mean number of infections caused by a typical infected individual in totally and partially susceptible populations, respectively (*Anderson and May, 1992*). However, individual variations from this mean can be of vital importance (*Lloyd-Smith et al., 2005*); a study of SARS-CoV-2 in Hong Kong suggested that 80% of transmission events resulted from only 19% of cases (*Adam et al., 2020*). Superspreader events are widely reported to play a key role in the spread of the virus in community settings (*Shen et al., 2004*; *Kucharski and Althaus, 2015*; *Hamner, 2020*; *Ebrahim and Memish, 2020*; *Lemieux et al., 2020*).

Transmission within hospitals has been identified as a critical concern in managing the COVID-19 pandemic (*Iacobucci, 2020*). Studying transmission in the hospital environment requires care in distinguishing cases acquired in the community from cases of nosocomial infection (*Sikkema et al., 2020*). The identification of outbreaks may be complicated by the potential for asymptomatic carriage of the virus (*Rivett et al., 2020*). As such, testing of asymptomatic health care workers (HCW) has been proposed as a means to reduce viral spread and protect the workforce and patients (*Black et al., 2020*; *Jones et al., 2020*).

Evaluating SARS-CoV-2 transmission in a hospital context is not a trivial task. Factors such as the date of symptom onset relative to the date of admission can be used to identify cases of likely nosocomial transmission (*Rickman et al., 2021*; *Price et al., 2021*). Phylogenetic methods can be used to identify putative clusters of infection occurring within a single ward or other location within a hospital setting (*Meredith et al., 2020*). Epidemiological methods can be used to look at potential contacts and opportunities for transmission between cases of infection (Cluster Track, Camart Ltd, Cambridge, UK). However, these approaches do not always provide the detail of who infected whom within a single outbreak or cluster, lacking the resolution to resolve the fine detail of transmission clusters.

Viral genome sequences provide a valuable resource for evaluating nosocomial transmission. At the most basic level, highly distinct sequences are unlikely to be related via transmission. Multiple approaches have been proposed to infer transmission patterns from genome sequences. Typically, phylogenetic reconstruction is used to infer relationships between sequences, an evolutionary model being combined with epidemiological data to infer a network of transmission events (*Volz and Frost, 2013*; *Ypma et al., 2012*; *Didelot et al., 2014*; *Hall et al., 2015*). Modelling approaches have been extended to include factors such as unsampled hosts (*De Maio et al., 2016*), the availability of multiple samples per patient (*Wymant et al., 2018*; *Worby et al., 2016*), incomplete epidemics (*Didelot et al., 2017*), and deep sequence data (*Ratmann et al., 2019*).

Here, we evaluated patterns of viral transmission occurring in epidemiological clusters in Cambridge University Hospitals NHS Foundation Trust (CUH), United Kingdom, where multiple patients with suspected hospital-acquired COVID-19 infections and/or HCW working on the same wards tested positive for SARS-CoV-2 within a 2-week period. Using a novel approach to combine genetic and epidemiological data, we inferred networks of SARS-CoV-2 transmission between these individuals. The tight clustering of genome sequences collected within a single ward places an imperative on the exploitation of non-genetic information to identify potential transmission events. We did this by combining an evolutionary model with symptom and location data for individuals considered, and knowledge of SARS-CoV-2 infection dynamics. Examining data from the largest clusters of infection identified within the hospital, we showed that the spread of infection in these clusters was driven by a small set of superspreader individuals.

**eLife digest** The COVID-19 pandemic, caused by the SARS-CoV-2 virus, presents a global public health challenge. Hospitals have been at the forefront of this battle, treating large numbers of sick patients over several waves of infection. Finding ways to manage the spread of the virus in hospitals is key to protecting vulnerable patients and workers, while keeping hospitals running, but to generate effective infection control, researchers must understand how SARS-CoV-2 spreads.

A range of factors make studying the transmission of SARS-CoV-2 in hospitals tricky. For instance, some people do not present any symptoms, and, amongst those who do, it can be difficult to determine whether they caught the virus in the hospital or somewhere else. However, comparing the genetic information of the SARS-CoV-2 virus from different people in a hospital could allow scientists to understand how it spreads.

Samples of the genetic material of SARS-CoV-2 can be obtained by swabbing infected individuals. If the genetic sequences of two samples are very different, it is unlikely that the individuals who provided the samples transmitted the virus to one another. Illingworth, Hamilton et al. used this information, along with other data about how SARS-CoV-2 is transmitted, to develop an algorithm that can determine how the virus spreads from person to person in different hospital wards.

To build their algorithm, Illingworth, Hamilton et al. collected SARS-CoV-2 genetic data from patients and staff in a hospital, and combined it with information about how SARS-CoV-2 spreads and how these people moved in the hospital . The algorithm showed that, for the most part, patients were infected by other patients (20 out of 22 cases), while staff were infected equally by patients and staff. By further probing these data, Illingworth, Hamilton et al. revealed that 80% of hospital-acquired infections were caused by a group of just 21% of individuals in the study, identifying a 'superspreader' pattern.

These findings may help to inform SARS-CoV-2 infection control measures to reduce spread within hospitals, and could potentially be used to improve infection control in other contexts.

## Results

We developed a method to infer networks of transmission events between individuals within CUH. Our method combines knowledge of SARS-CoV-2 infection dynamics with viral genome sequences and data describing the movements of patients and HCWs within the hospital.

Applying our method to these data, we generated maximum likelihood reconstructions of the pattern of transmission events occurring within five infection clusters, each of which was centred on a ward at CUH. For reasons of data protection we term these wards A to E. These five wards were chosen as they contained the largest number of patients with hospital-onset infections and/or health-care worker infections in CUH up to the end of the study period. Of the wards analyzed, wards A to D were 'green' wards (designated for patients who had not tested positive for SARS-CoV-2), while ward E was a 'red' ward (designated for known COVID-19 patients). Although referred to here as a 'ward' for simplicity, one of the green wards was a number of neighbouring clinical areas within the hospital. A preliminary analysis of the data, treating individuals in a pairwise manner, suggested that transmission events between the identified wards was unlikely (*Figure 1*).

Reconstructed transmission networks for the four green wards are shown in *Figure 2*. Our method requires each transmission in a network to be consistent with a statistical model of pairwise viral transmission (*Illingworth, 2020*). As such, a broad range of possibilities could in theory be inferred. At one extreme, the infections on a ward could all arise from a single introduction of the virus, with all cases arising via transmission from a single individual. At the other extreme, the infections could all be entirely independent of one another, with no transmission between cases at all. Our approach uses sequence data and epidemiological information to identify cases that are plausibly linked by direct transmission, before inferring the maximum likelihood network reconstruction of events. Across the green wards, the majority of cases were inferred to be connected to at least one other via transmission, with 42 out of 54 cases being joined into networks. These networks involved between 2 and 11 cases each (mean 5.9 cases). This contrasts with results from the single red ward (*Figure 3*), in which only 9 of 19 cases were inferred to be linked to others via transmission (mean

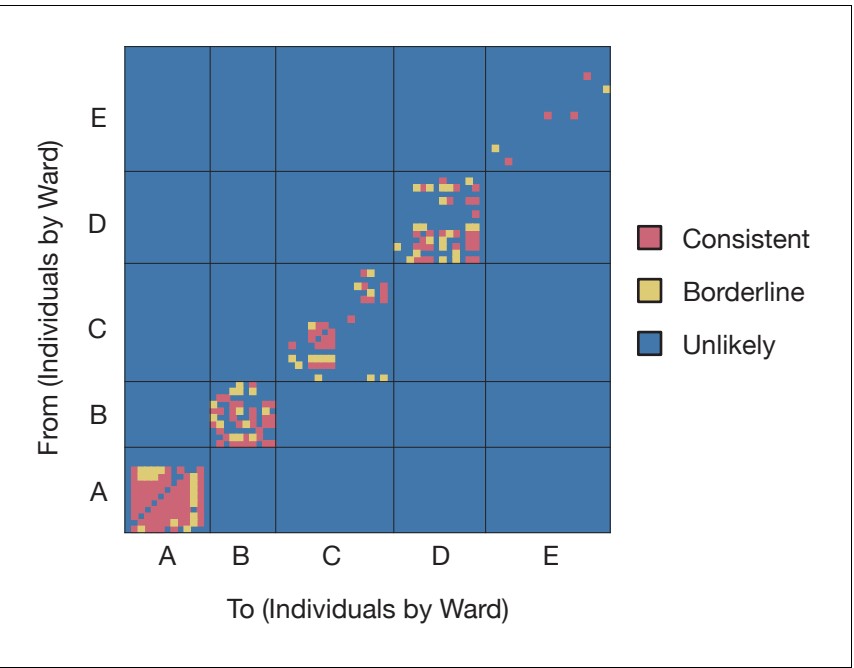

**Figure 1.** Preliminary analysis of the data with A2B-Covid. Squares indicate the extent to which an individual-to-individual transmission event is consistent with the data collected, when considered on a pairwise level. Our analysis highlighted multiple potential transmission events occurring within each ward, but transmission between individuals on different wards was uniformly assessed as unlikely. Further analyses of the data considered wards as independent and isolated locations.

The online version of this article includes the following source data for figure 1:

**Source data 1.** Assessment of pairwise transmission events.

inferred network size 2.3 cases). Our result corresponds to the nature of the wards studied; the repeated transfer of infected patients onto a red ward leads to an increased number of independent introductions.

Individuals in our study were divided into patients and HCWs allowing for the estimation of rates of transmission between these categories. Of the 38 transmission events in the maximum likelihood networks, 20 were patient-to-patient, 8 were from patient to HCW, 8 were HCW-to-HCW, and just 2 were from HCW to patient (*Figure 2—figure supplement 1*) These results suggest that patients were significantly more likely to be infected by other patients than by health care workers (p-value $6.1 \times 10^{-5}$, one-tailed binomial test). By contrast, HCWs were at approximately equal risk of being infected by patients and other HCWs.

Some of the wards analysed appeared to show uneven patterns of viral transmission, with a small number of individuals responsible for most of the infections observed. For example, in the maximum likelihood reconstruction derived for ward A, the majority of individuals infected did not pass on the virus, while individuals A6 and A10 were the sources, respectively, of four and five transmission events (*Figure 2A*).

A statistical analysis of the inferred networks provided evidence for a role for superspreading behaviour during transmission. As a first step in evaluating this, we calculated the level of uncertainty in each inferred network, combining data from the maximum likelihood network with that from other plausible, but lower-likelihood networks. *Figure 2—figure supplement 2* shows statistical ensembles of networks inferred for the green wards. In this figure, the width of an arrow is proportional to the probability that transmission occurred between each pair of individuals. The maximum likelihood network inferred for Ward E was the only plausible solution given by our reconstruction method.

In a second step, we fitted models to data from these ensembles, calculating probability distributions of the number of individuals infected by each person in the dataset. A negative binomial

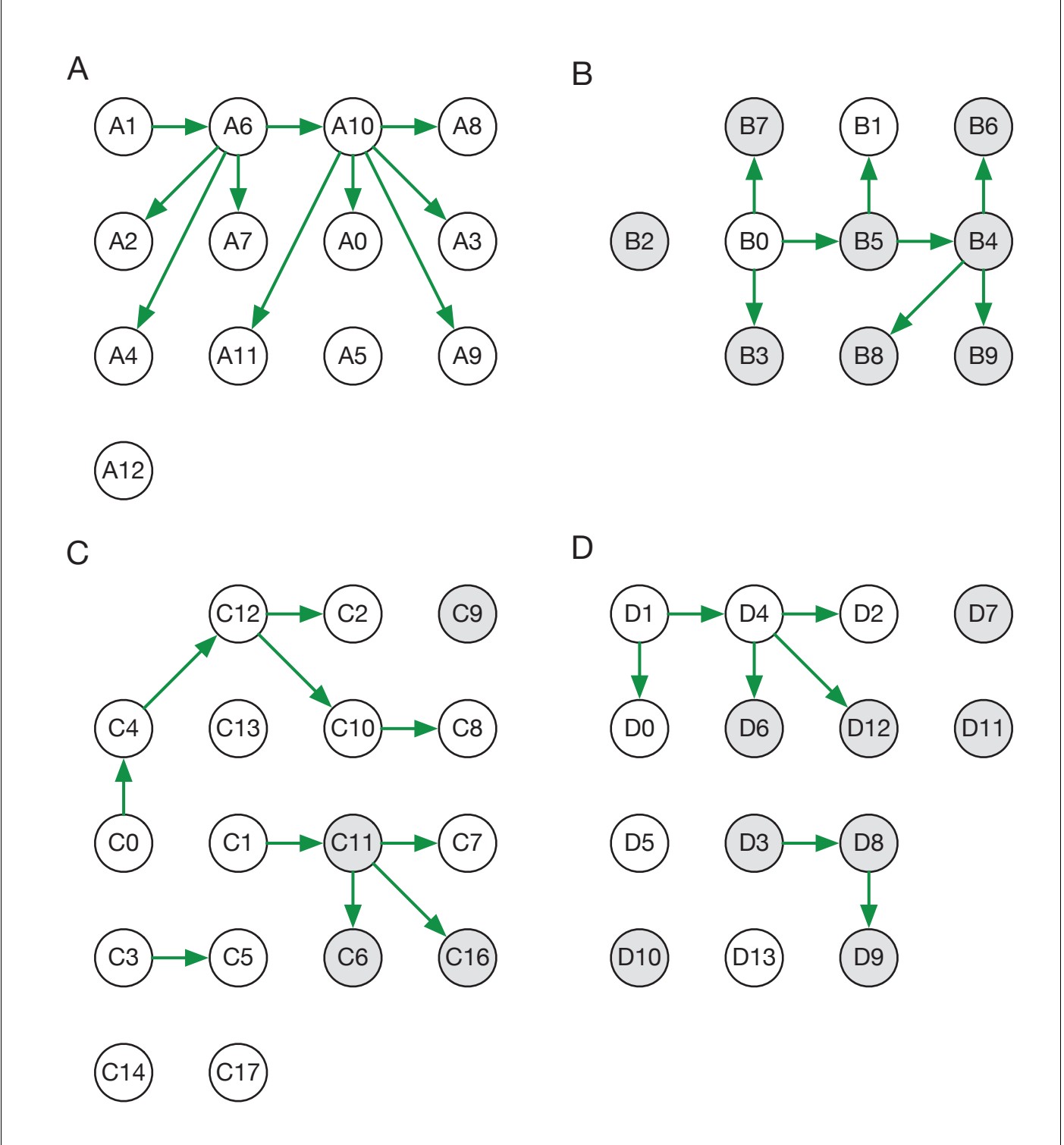

**Figure 2.** Maximum likelihood transmission networks for wards A to D. Circles represent individuals and arrows show transmission events. White circles represent patients while grey circles represent health care workers. Individuals for which no transmission events were inferred are represented as isolated circles.

The online version of this article includes the following source data and figure supplement(s) for figure 2:

**Figure supplement 1.** Maximum likelihood sources of patient and HCW infections.

**Figure supplement 2.** Ensemble transmission networks for wards A to D.

*Figure 2 continued on next page*

*Figure 2 continued*

**Figure supplement 2—source data 1.** Posterior probabilities of transmission between individuals on each ward.

model, in which the extent of viral spreading was overdispersed, gave a better explanation of the data, measured using the Bayesian Information Criterion, than a simpler model in which all individuals transmitted equally (*Figure 4*). In the best-performing model of viral spreading, 87% of individuals either did not transmit the virus, or transmitted only to one other. Taken across all individuals, 21% of individuals were responsible for 80% of viral transmission, a result very similar to that found among the general population (*Adam et al., 2020*). A repeat of this calculation for the green wards alone gave similar statistics (*Figure 4—figure supplement 1*), with 23% of individuals in these wards being responsible for 80% of transmission (*Figure 4—figure supplement 1*).

In our maximum likelihood reconstructions, a total of five individuals infected three or more others, including one HCW and four patients. Clinical characteristics of these individuals were explored, but are not described in detail or assigned to their anonymised ward clusters to preserve patient anonymity. Of the four patients, all had suspected hospital-acquired COVID-19 and significant comorbidities: two had a history of chronic liver disease, and two had previous haematological malignancies, one of whom was still on immunosuppressive treatment. Immunosuppression has been associated with prolonged viral shedding (*Italiano et al., 2020*; *Avanzato et al., 2020*). One superspreader was confused and mobile on the ward. Another had a fever for several days before being tested for SARS-CoV-2, which had been attributed to a pre-existing community-acquired bacterial infection. The only HCW superspreader exclusively infected other HCWs, and shared accommodation with several of these individuals. Cycle threshold (Ct) values of samples collected from identified superspreader individuals were not statistically distinct from those from individuals in the study in general (*Figure 4—figure supplement 2*).

Inferred timings of transmission events caused by superspreaders showed a variety of patterns (*Figure 4—figure supplements 3–5*). In ward B, the initial three infections of HCWs by the individual B0 are likely to have occurred within a short period of the SARS-CoV-2 virus entering the ward (within 4 days with 95% certainty), suggesting that an outbreak caused by superspreading may

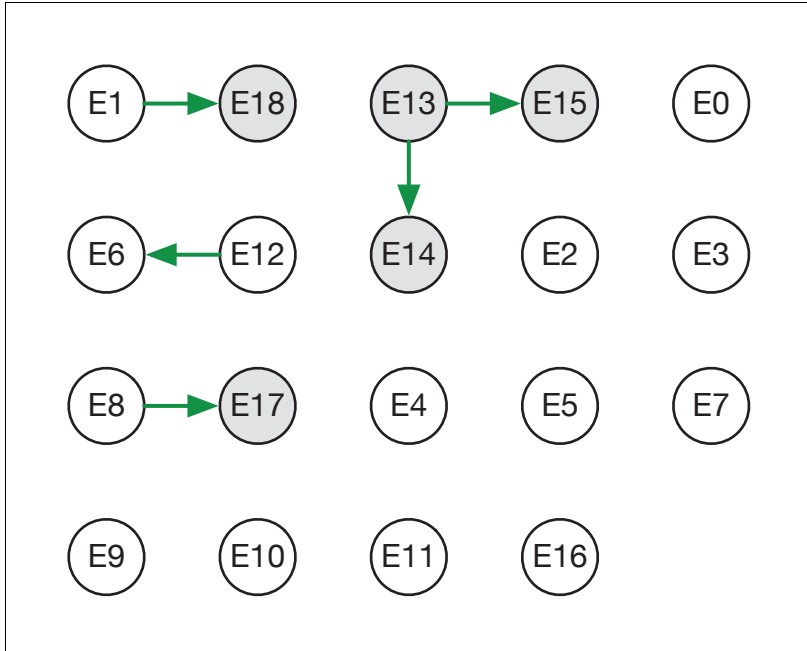

**Figure 3.** Maximum likelihood transmission network for ward E. Circles represent individuals and arrows show transmission events. White circles represent patients while grey circles represent health care workers. Individuals for which no transmission events were inferred are represented as isolated circles.

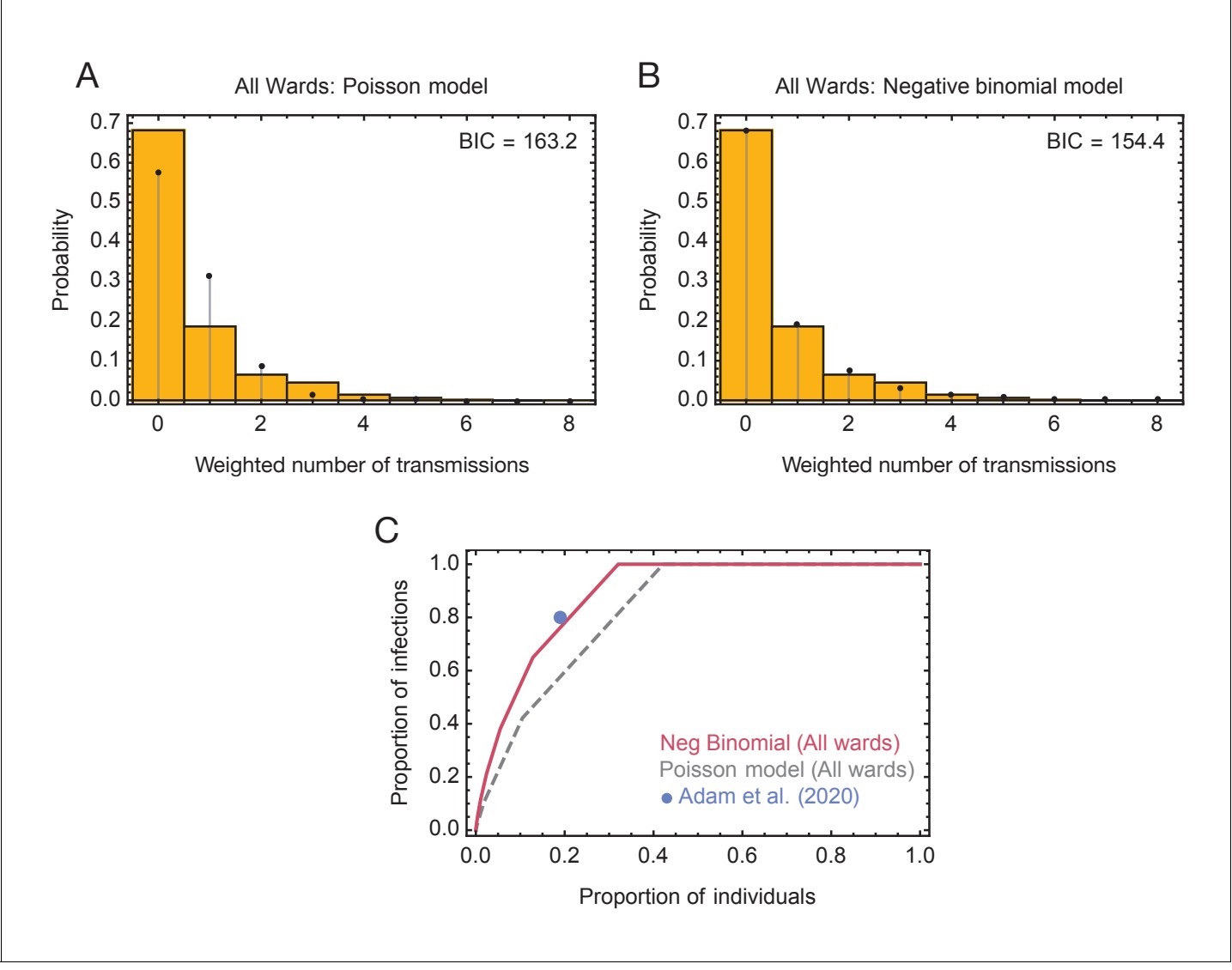

**Figure 4.** Models of viral transmission. (**A**) Fit of the output of the Poisson model (black dots) to the ensemble data (yellow bars). The weighted number of transmissions per individual reflects the uncertainty in the network reconstruction across the ensemble. (**B**) Fit of the output of the negative binomial model (black dots) to the ensemble data (yellow bars). (**C**) Proportions of individuals causing different proportions of infections. A negative binomial model (red line) fitted to all ward data produces a result similar to that of *Adam et al., 2020* (blue dot), with 20% of individuals being responsible for 80% of infections. A Poisson model fitted to the same data (dashed grey line) has 20% of individuals being responsible for 60% of infections.

The online version of this article includes the following source data and figure supplement(s) for figure 4:

**Source data 1.** Distributions of number of individuals infected by each individual and fits to these data using Poisson and Negative Binomial models.

**Figure supplement 1.** Modelling of viral transmission on the green wards.

**Figure supplement 1—source data 1.** Distributions of number of individuals on green wards infected by each individual and fits to these data using Poisson and Negative Binomial models.

**Figure supplement 2.** Ct values of viral samples.

**Figure supplement 3.** Inferred timings of transmission events in ward A.

**Figure supplement 3—source data 1.** Distributions of timings of transmission events on ward A.

**Figure supplement 4.** Inferred timings of transmission events in ward A.

**Figure supplement 4—source data 1.** Distributions of timings of transmission events on ward B.

**Figure supplement 5.** Inferred timings of transmission events in ward D.

**Figure supplement 5—source data 1.** Distributions of timings of transmission events on ward D.

spread rapidly to multiple individuals. However, the inferred timings on other wards were less conclusive; in wards A and C the inferred distributions of infection times were more diffuse. We note simply that where superspreader events occur, the potential exists for multiple transmissions to occur within a short space of time.

## Discussion

We have here outlined a novel approach for the inference of transmission networks. Our approach combines an evolutionary model with specific information about SARS-CoV-2 transmission dynamics to identify the most probable set of transmission events linking a set of cases of infection. Our approach builds upon previous approaches to analysing SARS-CoV-2 data from hospital settings, going beyond the identification of clusters to infer directional networks of viral transmission.

The multiple forms of data used by our method each play a critical role in network inference. Where individuals in a ward become symptomatic at similar times, sequence data provide a strong indication of whether these infections are linked via transmission or arise from completely independent events. However, the potentially short time spanned by a local outbreak may be insufficient for substantial genetic variation to accumulate in the virtual population; in such cases, other information, such as dates of symptom onset, become critical for network reconstruction.

The inference of networks allows for detailed study of how SARS-CoV-2 can spread within a clinical environment. Contacts between patients appear to be crucial, as they are primarily infected by other patients rather than through transmission from HCWs. This finding has potential implications for the application of protective measures within a hospital environment, whereade face mask usage was enforced for individuals in outpatients and for HCWs in all areas of the hospital, inpatients were not at the time of data collection subject to the same precautions. A recent study has suggested SARS-CoV-2 aerosolisation to be high in areas where patients with COVID-19 are coughing (*Hamilton, 2021*).

Our study is biased in its consideration of the largest clusters of infection identified in CUH wards during the first wave of the pandemic. Examining data from these clusters, we identified a pattern of superspreading, in which a small proportion of individuals were responsible for the majority of nosocomial transmission events. Our result is interesting in the context of previous studies of superspreading (*Shen et al., 2004*; *Kucharski and Althaus, 2015*), providing an example of this in a hospital context, and a case in which a small proportion of 'superspreader individuals' drive 'superspreader events'. We note that, while prolonged or increased viral shedding would increase the chance of an individual becoming a superspreader, behavioural and environmental factors may also be influential.

A key feature of SARS-CoV-2 that makes infection prevention and control (IPC) particularly challenging is its significant infectivity prior to the onset of symptoms. This means that isolating staff or patients once symptoms are recognised is not sufficient to prevent transmission. The superspreaders identified here illustrate several principles for limiting the spread of SARS-CoV-2 in hospitals. First, scrupulous adherence to infection control practices including use of appropriate personal protective equipment (PPE) at all times, even on green wards and in non-clinical hospital areas, is required to limit transmission between asymptomatic patients and staff in which COVID-19 is not suspected. Use of masks by patients, including on green wards, should be instituted if tolerated, particularly when staff are present in patient bed spaces. Second, healthcare professionals must be vigilant to the possibility of hospital-onset COVID-19 and have a low threshold for testing inpatients, even where an alternative differential diagnosis for the patients' symptoms exists. Third, as soon as positive cases are confirmed, appropriate isolation and PPE precautions should be used, along with contact tracing, testing and isolation. Patients who have been in direct contact with confirmed cases on green wards should be isolated. Fourth, regular screening of asymptomatic individuals can help to identify patients and staff that may be unsuspectingly infected with COVID-19 and infectious, either presymptomatic, pauci-symptomatic or asymptomatic, prompting isolation and contact tracing (*Jones et al., 2020*). Fifth, ventilation should be improved to reduce the risk of aerosol dispersal. Of note, our recommendations concur with the current UK guidance for COVID-19 infection prevention and control (*Public Health England, 2020*), which have evolved during the course of the pandemic. Finally, our identification of transmission from patients to HCWs highlights the use of higher grade

respiratory precaution to protect HCWs (such as FFP3 respirators) as an important topic for future research.

The potential for superspreading enhances the difficulty of controlling hospital-acquired infection, particularly as most transmission events from superspreaders to other people inferred in this study occurred within a relatively short time period. By the time a second linked case in a ward is identified, the potential exists for an index case to have infected multiple individuals, making it too late to prevent a broader outbreak. While this study does not allow for a complete characterisation of superspreading individuals, it may suggest possible risk factors in these instances such as immunosuppression (associated with prolonged shedding), more mobile behaviour that may have contributed to increased risk of transmission, and extended symptoms (fever) prior to testing and isolation (due to fever being attributed to an alternative cause).

Our inferences of transmission were performed on the basis of a largely complete dataset. Sampling of infections within wards was likely very close to complete, with sample collection from symptomatic patients and health care workers being conducted in parallel to asymptomatic screening of hospital staff. A screening programme for asymptomatic HCW was set up at CUH in April 2020 (*Rivett et al., 2020*) and voluntary weekly screening is currently offered to all HCW. SARS-CoV-2 seroprevalence among staff tested in CUH between 10th June and 7th August 2020 was 7.2% (*Cooper, 2020*). The five outbreaks described here occurred earlier in the pandemic (March to June 2020), when staff seroprevalence would have been lower. The proportion of staff with neutralising antibodies during the outbreaks was therefore low, and likely played a minor role in transmission dynamics. Sequencing was attempted for all positive samples; across the green wards data was of high quality for 55 out of 71 individuals for whom data was collected (>80% unambiguous nucleotides with no more than one ambiguous nucleotide at at a variant site) consensus viral genome in addition to data describing their location during the period of the study.

We acknowledge several limitations to our study. There is potential for missing or incomplete data, with some aspects of the data more vulnerable than others to omission. Asymptomatic screening was offered to all staff working on the five wards analysed in the study during the outbreaks. It is theoretically possible that HCW could have caught COVID-19 early on in the outbreaks and cleared the virus quickly, becoming negative at time of testing, or caught the virus asymptomatically after the screening test, or had levels of virus below the detection limit of the assay (and thus have been false negatives). However, levels of SARS-CoV-2 RNA below the assay detection limit are unlikely to be infectious (and thus not significant for the inferred transmission networks), and overall HCW testing coverage was high. Testing of asymptomatic patients varied by ward. Asymptomatic screening was done for all patients on Wards A and B during the outbreaks, and all patients entering Ward E (the only 'red' ward included in the study) were known SARS-CoV-2 positives. However, for Wards C and D, systematic asymptomatic screening of all patients on the ward during the outbreaks was not performed, and it is possible some asymptomatic infections (that could have contributed to the transmission networks) were missed. Data describing the wards on which patients were treated is likely to be complete, but the same ward data for HCWs may miss the potential for interactions between workers outside of their base wards for example in communal non-clinical areas within the hospital. Missing location data would lead to the non-identification of genuine contacts; our approach may therefore underestimate the number of infections caused by transmission between health care workers. Where data were missing our method does not attempt to identify cases of indirect transmission, for example invoking the presence of unobserved individuals. Only potential cases of transmission that were compatible with a model of direct transmission were included in our networks. The number of superspreader individuals identified in this study (five) is too small to draw general conclusions on superspreader characteristics. Moreover, it is not possible to disentangle whether superspreading was driven mainly by individual factors (such as infectivity or behaviour) or environmental factors (such as patient placement and ventilation at time of peak infectivity), or a combination of these. Ct values can vary for many reasons including the timing of sampling during COVID-19 infection, sampling type and technique, viral transport, sample preparation and variability between PCR runs. The finding that Ct values did not vary significantly between superspreader and non-superspreader individuals should therefore be interpreted with caution.

In conclusion, we have here applied a combined statistical approach to infer and examine SARS-CoV-2 transmission networks within a hospital environment during the first wave of the pandemic in the United Kingdom. For the largest ward outbreaks of hospital-onset COVID-19, the majority of

transmission was driven by a small proportion of individuals. Future developments could include exploring the impact of variables that may be associated with an increased transmission risk. Examples would include novel SARS-CoV-2 variants such as B1.1.7 and B1.617.2, which appear to be more readily transmissible (*Rambaut et al., 2020*), patient characteristics such as immunosuppression which are associated with prolonged viral shedding (*Avanzato et al., 2020*) and environmental factors such as patient placement and room ventilation. Nevertheless, this unusually comprehensive dataset has provided detailed insights into the processes of hospital-based transmission. Combining data from multiple sources into a single analysis provides increased resolution and insight into the pervasive problem of nosocomial viral transmission.

## Materials and methods

### Study design, setting, and participants

Prospective surveillance studies of COVID-19 infection in patients and healthcare workers (HCW) were conducted at Cambridge University Hospitals NHS Foundation Trust (CUH), as previously described (*Rivett et al., 2020*; *Meredith et al., 2020*). Nasopharyngeal swab samples were collected and submitted to the Public Health England (PHE) Clinical Microbiology and Public Health Laboratory (CMPHL) or the Department of Medicine, University of Cambridge for SARS-CoV-2 diagnostic testing, as detailed below. Samples included in this study were collected during the first epidemic wave, between 22nd March and 14th June 2020. Clinical, laboratory, and patient location data were extracted from the hospital information system (EPIC Systems Corporation, Verona, USA). HCW ward location data were collected by members of the HCW screening team. PHE recommendations for COVID-19 infection prevention and control, including PPE use, were followed throughout the course of the study.

### Testing criteria

Patients and HCW had separate testing criteria and sample workflows. HCW were tested in the CUH HCW screening programme, which included both asymptomatic screening and symptomatic testing arms. Asymptomatic screening at the time of this study was focused on staff working on COVID-19 'red' wards (designated for patients with confirmed COVID-19 infection), wards with hospital-acquired infection outbreaks, and wards with high rates of staff positivity. Suggested symptomatology to prompt staff testing are described in *Rivett et al., 2020*, divided into 'major' criteria (e.g. fever and/or new persistent cough) and 'minor' criteria (e.g. coryzal symptoms, headache, myalgia).

For all five of the outbreaks described in this study, all staff working on the outbreak wards were invited for screening by the CUH HCW screening team (i.e. tested regardless of any symptoms or if asymptomatic) during the outbreak periods (prompted by the outbreak investigations and/or high rates of staff positivity on the wards).

There was no systematic asymptomatic screening for inpatients in CUH during the study period, but targeted patient screening on wards with hospital-onset COVID-19 outbreaks was performed. Ward E was a COVID-19 'red' ward; all patients on this ward had tested positive prior to placement there. Wards A to D were all 'green' wards (designated for non-COVID-19 patients) at the time the hospital-onset COVID-19 outbreaks started. For Ward A, symptomatic contacts of confirmed cases were tested initially, and as the outbreak grew and more cases were confirmed, ultimately all patients on the ward were screened (including asymptomatics). For Ward C, contacts of confirmed positives and/or patients who developed symptoms were screened, and for Ward D, symptomatic contacts of the index case were tested. Thus, for Wards C and D, systematic asymptomatic screening of all patients on the ward during the outbreaks was not performed. For the Ward B outbreak, when the index patient (case B0) tested positive, all staff members who had worked on the cluster of clinical areas referred to as 'Ward B' within the preceding 2 weeks plus all patients on those wards were screened (regardless of any symptoms). Thus, there is high confidence for Wards A, B and E that all infections among both staff and patients were detected (providing the amount of SARS-CoV-2 RNA was sufficient for detection).

Visitor restrictions were introduced on 25th March 2020 and so were present for almost all of this study (first positive swab was for Ward E, collected a few days before this). After 25th March, visitors

to adult patients were only permitted in exceptional circumstances: for patients at the end of life or visitors with a direct care role for the patient.

## Laboratory methods

Samples underwent nucleic acid extraction and were tested for presence of SARS-CoV-2 using a validated in-house RT qPCR assay, as previously described (*Price et al., 2021*). The test was reported as SARS-CoV-2 PCR positive if the cycle threshold (Ct) value was less than or equal 36. A 15 microlitre aliquot of the RNA extract of each positive sample was transferred to the Department of Pathology, University of Cambridge, for sequencing.

## Sequencing and genomic analysis

Samples were assigned COG-UK sequencing codes and sequenced using a multiplex PCR based approach according to the modified ARTIC v2 protocol with v3 primer set (*artic-ncov, 2019*; *Quick, 2020*). Amplicon libraries were sequenced using MinION flow cells v9.4.1 (Oxford Nanopore Technologies, Oxford, UK). Genomes were assembled using reference-based assembly and a bioinformatic pipeline (*Artic Network, 2021*). All sequences underwent quality control (QC) filtering, including a 20x minimum coverage cut-off for any region of the genome and 50.1% cut-off for calling single nucleotide polymorphisms (SNP).

## Sample selection

Patients from CUH were determined to have indeterminate, suspected or definite hospital-acquired infections (HAI) on the basis of days from admission to first positive SARS-CoV-2 test, using the same definitions as in *Meredith et al., 2020*; *Price et al., 2021*: indeterminate = positive test after 48 hr and less than 7 days post admission; suspected = positive test 7 to 14 days post admission; definite = positive test greater than 14 days post admission. Wards were ranked by their combined number of indeterminate, suspected and definite HAI cases plus HCW cases (taking the HCW ward to be any ward each HCW had worked on within the preceding 14 days before testing positive). 15 wards at CUH had two or more HAI and HCW cases occur within 14 days of each-other. The five wards with the largest number of combined HAI and HCW cases were chosen for this study and named wards A to E (*Supplementary file 1*).

Each of the selected wards had 10 or more HAI plus HCW cases, therefore representing the largest HAI ward-based clusters in CUH during the study period, which encompasses the 'first wave' of the pandemic in the East of England region. Four were 'green' wards (A to D), intended to house patients who did not have COVID-19, and one was a 'red' ward (E), intended to house confirmed COVID-19 cases. In four out of five wards, the majority of cases were HCW. Ward characteristics cannot be described in detail in order to preserve confidentiality. They were typically organised into bays, with four to six patients per bay, and a limited number of side rooms (which are critical for infection control purposes). In summary, Ward A had 30 beds with three side rooms; Ward C had 27 beds with three side rooms; Ward D had 30 beds with four side rooms; Ward E had 26 beds with three side rooms. The Ward B outbreak was focused around a ward with 32 beds of which 12 were side rooms, although several adjacent clinical areas were screened as part of the outbreak investigation (as staff were shared between these wards).

An initial network analysis of patient bed movements was conducted to add patients to each ward cluster that may have been in contact with the HAI cases, either with community onset infections on the same ward or while they were co-located on other wards outside of the 'outbreak wards' themselves. This analysis was undertaken in two steps using SQL v18.5.1 and FoodChain-Lab (an extension of the Knime Analytics Platform v3.6.1). The first step involved utilising SQL to process case and ward movements data from CUH patients, creating a list of ward-based case co-locations that were within the set parameters of the network analysis model. These parameters were (1) an infectious period that included the 4 days prior to symptom onset up until 7 days after symptom onset, and (2) a susceptibility period of 14 days prior to symptom onset. Where symptom onset date was not available, the case positive specimen date was used instead. The second step was to import the case co-locations data into FoodChain-Lab, which was used to draw a social network diagram of cases and their ward co-locations with one another that met the set parameters. This network diagram was used to identify any clustering of cases by ward that had not met the original criteria of

HAI and HCW cases but could have been involved in shared transmission with those individuals based on their co-location within the infection period of the virus. This yielded the final case set for each ward cluster taken forward for analysis using the transmission reconstruction model.

## Data collection

Data were collected on COVID-19 symptom onset dates for all included individuals from the five ward clusters. These were collected separately for HCW and patients. HCW testing was part of the CUH HCW screening programme, and testing criteria are described in Rivett et al (*Sikkema et al., 2020*). (*Table 1*). Staff who tested positive were then contacted by members of the HCW screening team and asked retrospectively about their symptoms, and onset dates were recorded by the HCW screening team and used in this analysis. For patients, symptom onset dates were collected by retrospective review of patients' electronic hospital records (EPIC Systems), usually noted at presentation by the clerking doctor but all records were examined for any suggestion of symptoms. Symptom definitions followed the standard national recommendations at the time: initially fever, breathlessness, and new continuous cough; muscle aches were added in early May and loss of taste or smell was included from mid-May. If the infection was asymptomatic, or symptom onset dates could not be determined, then the date of first positive SARS-CoV-2 test was used instead.

Patient ward movement data through the hospital was downloaded from the hospital electronic records system (EPIC). Data on HCW shift patterns were collected manually using hospital shift rostering information and, in some cases, directly contacting the HCW. HCW were defined as being either present or absent on each 'outbreak' ward within each 24 hr period (i.e. time of day or shift length was not taken into account). No HCW worked on multiple outbreak wards, although staff contact outside of wards (during lunch, outside of work etc) cannot be excluded.

Each anonymised patient code was linked to COG-UK sequencing codes, with sample collection date and laboratory receive dates recorded for each sample that was sequenced. Dates of symptom onset and sample collection for each individual are shown in *Figure 5*.

## Model development

Given a set of data from infected individuals, comprising viral genome sequences collected from each infection, dates on which individuals became symptomatic, and when individuals were co-located, we sought to identify whether and how these cases are linked by transmission. We noted that, prior to analysis, we did not know whether the cases for which we had data were connected via

**Table 1.** Case numbers in the five major ward clusters.

'Total cases before network analysis' were derived by adding patients with potential hospital-acquired COVID-19 infections and HCW cases from each ward. The five wards with the largest combined number of HAI and HCW cases within the study period were analysed, with anonymised ward names A to E. 'Ward type' refers to whether wards were 'green' (intended for patients negative for COVID-19), or 'red' (intended for COVID-19 patients). The breakdown of HAI and HCW cases for each of the included wards is shown in columns 'HAI cases before network analysis' and 'HCW cases before network analysis', respectively. The 'network analysis' at this stage identified additional patients that could have been involved in transmission with the HAI patients on the basis of co-location on the same or other wards within a plausible timeframe for SARS-CoV-2 transmission (described in Materials and methods). This yielded the final cases analysed for each ward cluster using the transmission reconstruction model. The final column shows the number of cases from each ward cluster for which genomic data were available. In total, there were 98 cases with genomic data and 129 SARS-CoV-2 genomes analysed in the study (the larger number of genomes than cases is because of multiple samples per patient that underwent SARS-CoV-2 sequencing). Three patients were included in two ward clusters each (which is why the total of the 'Cases after network analysis with genomic data' column is 101). HAI = hospital-acquired infection (definition in Methods); HCW = healthcare worker.

| Ward name | Ward type | Total cases before network analysis | HAI cases before network analysis | HCW cases before network analysis | Cases after network analysis | Cases after network analysis with genomic data |
|---|---|---|---|---|---|---|
| A | Green | 14 | 12 | 2 | 16 | 15 |
| B | Green | 11 | 2 | 9 | 15 | 12 |
| C | Green | 12 | 5 | 7 | 20 | 19 |
| D | Green | 14 | 4 | 10 | 16 | 16 |
| E | Red | 13 | 3 | 10 | 47 | 39 |

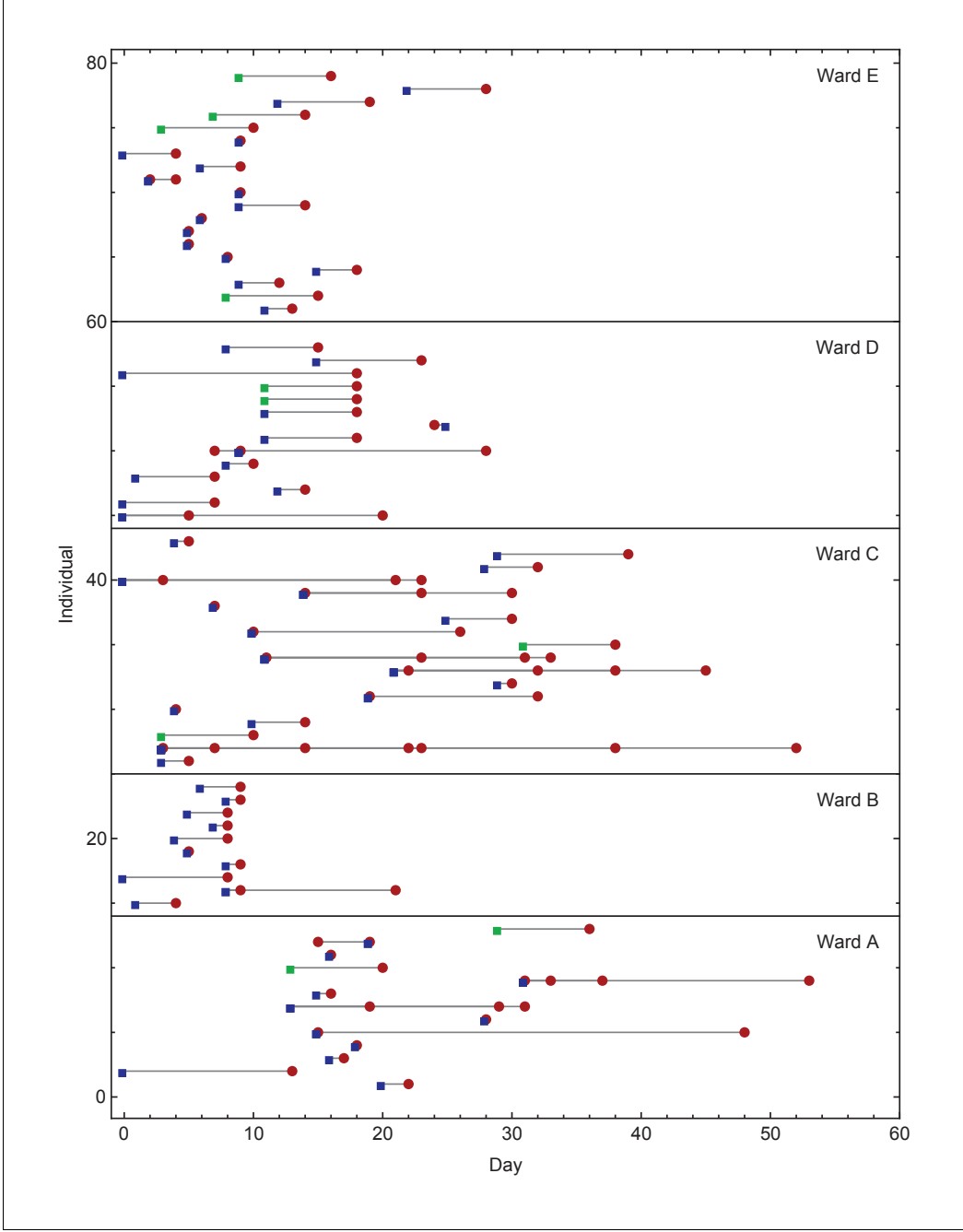

**Figure 5.** Overview of events on different wards. Blue squares show days on which individuals became symptomatic, while green squares show inferred days of individuals becoming symptomatic when these dates were unknown or not applicable. Red circles show days on which samples were collected from individuals for genome sequencing. Dates within each ward are normalised so that the first event on any ward is day zero. We note that not all collected samples led to genome sequences of sufficient quality to be useful in this study. The online version of this article includes the following source data and figure supplement(s) for figure 5:

**Figure supplement 1.** Ensemble transmission networks for wards B, D, and E generated without extending the times at which HCWs were present beyond the direct observations made.

**Figure supplement 1—source data 1.** Posterior probabilities of transmission between individuals on each ward inferred under a model in which no padding for HCW locations was included.

**Figure supplement 2.** Modelling of viral transmission in the absence of an extension to HCW locations.

*Figure 5 continued on next page*

*Figure 5 continued*

**Figure supplement 2—source data 1.** Distributions of number of individuals on all wards infected by each individual, as inferred under a model in which no padding for HCW locations was included, alongside fits to these data using Poisson and Negative Binomial models.

**Figure supplement 3.** Assigning mutations to the transmission tree.

**Figure supplement 4.** Restrictions placed on the network by sequence variants.

**Figure supplement 5.** Convergence of the statistical ensemble of networks for ward A.

**Figure supplement 5—source data 1.** Probabilities of transmission events between individuals inferred from data describing ward A calculated across partial and fuller sets of networks.

---

transmission; a set of cases may be anything from completely unlinked to a single connected transmission network.

As a first stage in our method, we partitioned our data into subsets of cases that might plausibly be linked by transmission events. This step was conducted using a pairwise measure of the consistency of the data observed from two individuals with the hypothesis of direct transmission having occurred between these two individuals; calculation of this measure was previously implemented in the A2B-COVID software package (*Illingworth, 2020*). The mathematical principles of this pairwise measure of consistency are explained fully elsewhere (*Illingworth, 2020*); below we provide a summary.

## Pairwise likelihood of transmission

Our pairwise likelihood function combines information from a broad variety of sources. Firstly, dates on which individuals first reported symptoms give an indication of the likelihood of transmission between individuals. Previous publications have characterised the infectivity profile of SARS-CoV-2 (i.e. the time between becoming symptomatic to transmitting the virus to another person conditional on causing infection), by fitting a shifted gamma distribution to data from confirmed infection, inferring the distribution parameters $\alpha=97.185$, $\beta=0.2689$, and shift $s=25.625$ (*Ashcroft et al., 2020*; *He et al., 2020*). Further, the time between an individual being infected with SARS-CoV-2 and developing symptoms has been described as a lognormal distribution, with inferred parameters $\mu=1.434$ and $\sigma=0.6612$ (*He et al., 2020*; *Li et al., 2020*). We use these distributions to assess the extent to which the observed symptom-onset times are compatible with transmission. Specifically, we denote by $X_T$ the event that transmission from i to j occurred on day T, and by X the event that transmission from i to j occurred at all. For convenience we collect together parameters, writing $\theta_1=\{\alpha, \beta, s, \mu, \sigma\}$. Supposing that individuals i and j became symptomatic on the days $S_i$ and $S_j$ respectively, we then derive the expression:

$$P(T|\theta_1,S_i,X)P(S_j|\theta_1,X_T) = \left[\int_{T-S_i-0.5}^{T-S_i+0.5} \frac{e^{-(x+s)/\beta}(x+s)^{\alpha-1}\beta^{-\alpha}}{\Gamma(\alpha)}dx\right]\left[\int_{S_j-T-0.5}^{S_j-T+0.5} \frac{e^{-\frac{(\log(x)-\mu)^2}{2\sigma^2}}}{x\sigma\sqrt{2\pi}}dx\right]$$

where a time of becoming symptomatic was unknown, the date at which that individual tested positive was used, corrected by a constant term estimated from the distribution of known times between positive test dates and dates of becoming symptomatic (*Illingworth, 2020*).

We elaborate on this model by incorporating data describing the location of individuals, noting that transmission can only occur when people are co-located, and viral genome sequences collected from individuals in the study.

## Incorporating location data

We have, for individuals in our study, information describing the wards on which patients were hosted, and the wards on which HCWs worked shifts. Given individuals i and j we defined a contact metric $w_{ij}(T)$ equal to the probability that i and j were co-located on any given day T.

Ward data describing patient locations and health care worker shift dates were used to calculate values $w_{ij}$ for each pair of individuals. For any given ward, we set the value $w_i(W,T)$ to equal one if the individual i was known to be on ward W on day T. If a healthcare worker worked a shift on ward W on day T, we further assigned a minimum value of 0.5 to the values $w_i(W,T-1)$ and $w_i(W,T+1)$, accounting for night shifts overlapping days, and the potential for fomite transmission. Removing

these additional days led to some small alterations in the inferred networks (*Figure 5—figure supplement 1*), but did not substantially affect our primary result, with again 80% of cases being caused by 21% of individuals (*Figure 5—figure supplement 2*). For all other dates, we set $w_i(W,T)$ to be zero. In the event that location data was unknown, we assigned a value for $w_i(W,T)$ of 1 for patients for the most common ward on which individuals in a cluster were located, and a value for $w_i(W,T)$ of 4/7 for health care workers, reflecting common shift patterns of work. The value $w_{ij}(T)$ for any two individuals i and j was then defined as the maximum of the product of $w_i(X,T)$ and $w_j(X,T)$ calculated across all wards X for that value of T.

## Incorporating viral genome sequences

Our pairwise likelihood accounted for data from viral genome sequencing using a separate likelihood function. For each pair of individuals, sequences were assessed according to the number of nucleotides by which they differed from a pairwise consensus, calculated from the two sequences, with reference to the broader set of sequences collected from a ward. At a given genome position, the consensus was defined as the nucleotide shared by the two sequences if they were identical, or by the most common nucleotide in the broader set if the two sequences differed.

The Hamming distances between each individual sequence and the pairwise consensus were measured. Each distance reflects both the potential evolution of the virus and the extent of measurement error inherent to the sequencing process. Using data from CUH, an estimate of the measurement error was previously calculated as E=0.414 nucleotides per pair of sequences, or 0.212 nucleotides per sequence (*Illingworth, 2020*). To model viral evolution we adopted the rate $\gamma_G$=0.0655 nucleotide substitutions per day, equal to the global rate of viral adaptation (*Hadfield et al., 2018*). We then derived a Poisson model for the observed number of nucleotide changes resulting from evolution and error. We denote the days on which viral sequence data was collected from the individuals i and j as D = {$D_i$, $D_j$}, and the Hamming distances of each sequence from their consensus as $H_i$ and $H_j$, and again use a shorthand expression $\theta_2$={E, $\gamma_G$}. We then have the result that

$$P\big(H_i,H_j|\theta_2,D,X_T\big) = \left(\frac{(E/2+\gamma_G P_i)^{H_i}e^{-(E/2+\gamma_G P_i)}}{H_i!}\right)\left(\frac{\big(E/2+\gamma_G(D_j-Q_i)\big)^{H_j}e^{-\big(E/2+\gamma_G(D_j-Q_i)\big)}}{H_j!}\right)$$

We now combine the above terms, writing y = {$S_i$, $S_j$, $H_i$, $H_j$}, $\theta$ = {$\theta_1$,$\theta_2$}, and denoting by X the event that transmission occurred from i to j at any time to obtain

$$P(y|\theta,D,X) = \sum_T P(T|S_i,\theta_1)P\big(S_j|\theta_1,X_T\big)w_{ij}(T)P\big(H_i,H_j|\theta_2,D,X_T\big)$$

This expression was used to assess the consistency of the data y from each pair of individuals with the hypothesis of transmission from i to j.

## Pairwise thresholds

The value $p(y|\theta,D,X)$ is defined over a discrete space of symptom times and sequence measurements. Calculating this over all feasible terms within this discrete space, we derived threshold values, conditional upon D, for which 95% and 99% of genuine transmission events would obtain values greater than the threshold. In this way, potential transmission events were classified into 'consistent' events (probability greater than the 95% threshold), 'borderline' events (probability between the 95% and 99% thresholds) and 'unlikely' events (probability below the 99% threshold).

The pairwise analysis, described to this point, was used to assess the data for potential between-ward transmission events.

## Inferring a maximum likelihood transmission network

As a first step in identifying transmission networks in our data, we clustered individuals using the pairwise thresholds. By way of notation we denote the transmission event from individual i to individual j as i→j.

Beginning with a single subset containing an arbitrary individual, we added an individual j to the subset S if for some individual i in S, either i→j or j→i has a likelihood that was 'consistent' or

'borderline' under the pairwise likelihood. If no such subset existed, j was placed in a new subset, repeating the process until all individuals were clustered into sets between which transmission events were unlikely.

If clustering identified individuals that were alone within a subset, these individuals were removed from the analysis at this point, giving further attention to clusters that contained at least two individuals.

## Network likelihood of transmission

Building upon our pairwise method, we calculated the likelihood of a transmission network. A transmission network must contain at least one transmission event. We consider the network N, comprising transmission events $i_1 \rightarrow j_1$, $i_2 \rightarrow j_2$, etc, in which the transmission even $i_k \rightarrow j_k$ occurs on day $T_k$. We here write $T^* = \{T_k\}$ to represent the set of all transmission times.

Similar to the calculation above, we consider symptom onset times, the locations of individuals, and sequence components, using these to derive an expression for a particular network of transmission events. Considering the non-genetic data, we make the assumption that the intrinsic dynamics of each transmission are independent of one another, so that the joint distribution of the observed symptom times $\{S_i\}$ given a particular network and set ot transmission times $\{X_{T_k}\}$, is

$$P(\{S_i\}|\theta_1, \{X_{T_k}\}) = \prod_k w_{i_k j_k}(T_k) P(T_k|\theta_1, S_{i_k}) P(S_{j_k}|\theta_1, X_{T_k})$$

For the sequence component of the likelihood, we consider the transmission tree defined by our network. Adopting an infinite sites model (**Kimura, 1969**), we assume that each observed nucleotide substitution occurred only once, with the reversion of substitutions being impossible. Further, we assumed that, while a substitution that was observed in only one sequence could have arisen from measurement error, substitutions that were observed in two or more sequences could not have resulted from error. Under these assumptions, we classified potential substitutions by the sequences they would be observed in, then adopted a Poisson likelihood model, comparing the periods of time in which sets of substitutions could have occurred to the numbers of substitutions that were observed in the data (**Felsenstein, 1981**).

We first describe this likelihood for a transmission network in which all times of transmission are known. From the sequence data, we identified nucleotide substitutions occurring in viral sequences relative to the consensus sequence. We denote the number of substitutions observed in a sequence or group of sequences I by $M_I$. Windows of time were then assigned to substitutions in reverse order. Diagrams illustrating this process are shown in **Figure 5—figure supplement 3**.

We first consider the final transmission event to occur (**Figure 5—figure supplement 3A**). We suppose that this transmission is A→B, occurring at time $t_{AB}$. It is clear that the time of sample collection $D_B$ occurs no earlier than $t_{AB}$; we divide the scenario into two possibilities.

Firstly, if $D_A \geq t_{AB}$, we have that substitutions observed only in individual B must have occurred in the $D_B - t_{AB}$ days between $t_{AB}$ and $D_B$, while substitutions observed only in A must have occurred in the $D_A - t_{AB}$ days between $t_{AB}$ and $D_A$. We note that substitutions observed only in single individuals are treated as potentially arising from either evolution or noise. We therefore calculated the likelihood of having observed $M_A$ substitutions from a Poisson distributed variable with expectation $E/2 + \gamma_G(D_A - t_{AB})$, and $M_B$ substitutions from a Poisson distributed variable with expectation $E/2 + \gamma_G(D_B - t_{AB})$.

Alternatively, if $D_A < t_{AB}$, then there exists a time P, equal to the latest of the set of times including $D_A$ and all other times of transmission $t_{AX}$ from A to an individual X other than B. We note that mutations observed only in B can then have occurred in the $D_B - P$ days of this interval, while mutations that were observed only in A can only have arisen through sequencing error. We therefore calculated the likelihood of having observed $M_A$ substitutions from a Poisson distributed variable with expectation $E/2 + \gamma_G(D_A - P)$, and $M_B$ substitutions from a Poisson distributed variable with expectation $E/2$. We note that the inclusion of a non-zero rate of sequencing error allows for A to transmit to B at a time after the observation of a viral consensus sequence from A that contains a substitution not observed in the viral consensus from B.

A similar logical process was carried for each transmission event working backwards in time. A more general case, representing a joining of two branches of the tree, is shown in **Figure 5—figure**

*supplement 1B*. We here consider the individuals A, B, C, and D, noting that these may not be terminal branches in the tree. That is, if a mutation occurs early in the infection of individual C, it will be observed both in the sequence collected from C and in sequences collected from individuals to which C later transmits the virus.

In this case, there are a variety of possibilities here to enumerate. If $D_A \geq t_{AD}$, then mutations which are observed precisely in A, D, and individuals downstream of A and D, must occur in the window between $t_{AB}$ and $t_{AD}$. If $D_A < t_{AD}$ but $D_A \geq t_{AB}$, then the same mutations must occur in the window between $t_{AB}$ and $D_A$.

Finally, if $D_A < t_{AB}$, it is impossible for mutations to be observed in precisely A, D, and individuals downstream of A and D. However, there exists a point P, equal to the latest of a set of times including $D_A$ and all other times of transmission from A to another individual prior to $t_{AB}$. In this scenario, substitutions occurring between P and $t_{AB}$ will be observed in B, C, D, and all individuals of these.

In all the above scenarios, mutations which are observed in B and C, plus individuals downstream of B and C, must occur in the window between $t_{AB}$ and the latter of $D_B$ and $t_{BC}$. In each case, we calculate the likelihood of having observed the given number of substitutions from a Poisson distributed variable with expectation given by the rate of evolution $\gamma_G$ multiplied by the time available for substitutions to occur, plus the measurement error E/2 where a variant was observed in only one sequence.

In this manner, we derived a phylogenetic hierarchy, dividing the transmission tree into sets of individuals that would be expected to share genetic substitutions; we denote these sets $I_a$, for a=0, 1, 2, etc. For each such set, we calculate the length of time in which these substitutions would have to occur; we denote these times $t_a$, counting the number of substitutions in each set, denoted $M_a$. We then have that

$$P(\{I_a\}, \{t_a\}, \{M_a\}|\theta_2) = \prod_a \left( \frac{(\delta_a E/2 + \gamma_G t_a)^{M_a} e^{-(\delta_a E/2 + \gamma_G t_a)}}{M_a!} \right)$$

where the term $\delta_a$ is equal to one if $I_a$ contains a single individual and 0 otherwise.

We now construct our final expression. We write G = $\{\{I_a\}, \{t_a\}, \{M_a\}\}$, and note that G, through the values $t_a$, is dependent on the times of transmission T* and the times of sample collection, which we denote more generally by D = $\{D_k\}$. Denoting by y the set of all data collected from the individuals in the network, we then have the result

$$P(y|\theta, D, X_N) = \sum_{T* = \{T_k\}} \left[ \prod_{k=1}^n \left[ w_{i_k j_k}(T_k) P(T_k|\theta_1, S_{i_k}) P(S_{j_k}|\theta_1, X_{T_k}) \right] P(G|\theta_2, D, X_{T*}) \right]$$

where $X_N$ is the event that the transmission events in the network occurred at some set of times. We convert this into a log likelihood, calculating the likelihood of a given network N:

$$\log L(N|y) = \log P(y|\theta, D, X_N)$$

We find the maximum likelihood network N, and using likelihoods calculated for multiple networks generate statistical ensembles of networks, to estimate, for example, the number of individuals infected by a specific person on a ward.

## Identifying plausible networks and orderings

The likelihood calculation above presupposes the existence of an ordered transmission network with times assigned to each transmission event. Calculations were performed in order to derive these networks.

Plausible networks describe sets of transmission events that could potentially describe a pattern of individuals within a subset. Four criteria were used to identify such networks. Firstly and secondly, networks had to be acyclic, and span all individuals in a subset. Thirdly, networks had to be likely, with each transmission event in the network having a pairwise likelihood that was classified as 'consistent' with the pairwise likelihood model. Where computational time allowed it, that is, for all but ward A, this criterion was relaxed to also consider transmission events that were classified as 'borderline'; details on computability are given later in the Methods. Fourthly, networks had to be consistent with the observed sequence data.

Consistency with sequence data can be described in terms of patterns of shared substitutions observed in sequences from different individuals. We consider a set $I_a$ of all individuals in the subset with viral sequences that share a substitution or set of substitutions. Under a maximum parsimony assumption that variants are gained only once and cannot revert, consistency requires that (i) there can only be one transmission event i→j with i ∉ $I_a$ and j ∈ $I_a$ and (ii) with the exception of this specific individual j in $I_a$, there can be no transmission events j→k with j ∈ $I_a$ and k ∉ $I_a$ (*Figure 5—figure supplement 4*). This has to apply for all such sets $I_a$.

For each subset of individuals created using the pairwise likelihood an exhaustive search of plausible networks was conducted. In the event that no such network was identified, the assumption that the network must span all individuals in the subset was gradually relaxed, reducing the number of individuals required successively by one until at least one plausible network was identified. In some cases, this relaxation led to the identification of multiple plausible networks involving non-overlapping sets of individuals; we return to this later in the Methods section.

Given a plausible network, we next considered orderings of the transmission events that it comprised. To achieve this, transmission events were given a nominal index. In an ordering the placement of the transmission $i_a$→$j_a$ before $i_b$→$j_b$ implied that $i_a$→$j_a$ occurred before or at an identical time to $i_b$→$j_b$ if it was true of the indices that a < b, but implied that $i_a$→$j_a$ occurred strictly before $i_b$→$j_b$ if a > b.

Plausible orderings were required to fulfil three criteria. Firstly, on a logical principle, $i_a$→$j_a$ must occur before $i_b$→$j_b$ if $j_a$ = $i_b$; an individual must be infected before transmitting the virus. We insisted that an individual who had received the virus via transmission could not themselves transmit until at least one day after they had been infected. Secondly, orderings had to fulfil characteristics imposed upon them by the likelihood function. If the first day on which the pairwise likelihood $L^P$ describing transmission between $i_b$ and $j_b$ was greater than zero was after the last day on which the equivalent likelihood for the transmission between $i_a$ and $j_a$ was greater than zero, it followed that $i_a$→$j_a$ must occur before $i_b$→$j_b$.

A third criterion was imposed by the existence of shared substitutions in the sequence data. Returning to the criteria imposed for network plausibility we suppose that for the set of sequences $I_a$ that contain shared substitutions we have identified the transmission event i→j for which i ∉ $I_a$ but j ∈ $I_a$, and that there exist a series of transmission events in the network j→k for this j. In this case, the last transmission for which k ∉ $I_a$ must occur before the first transmission for which k ∈ $I_a$, the substitution event necessarily occurring in individual j between these two times (*Figure 5—figure supplement 5*). All plausible orderings were stored for each plausible network; we note that the existence of a plausible network did not imply the existence of a plausible ordering of transmission events.

Potential times for transmission events were determined by the first and last time points for which each event had a non-zero pairwise symptomatic likelihood. These times, in addition to the ordering constraints, impose a combinatorial set of possible times at which transmission occurred.

Finally, the likelihood of a transmission network was calculated as the sum over all orderings of the sum over all sets of times of the timings-dependent network likelihood. Denoting the set of all orderings of transmission events as O, and the set of possible sets T* as T**, we have

$$\log L_n = \log \sum_O \sum_{T^{**}} L_n^{T^{**}}$$

## Maximum likelihood and uncertainty

Calculating likelihoods for plausible transmission networks, the maximum likelihood network was identified. Uncertainty about properties of the network, for example whether a network includes a particular transmission event, and the number of individuals infected by each individual, was quantified by Bayesian methods. Assuming a uniform prior over the space of possible networks, the posterior probability that each potential network is the true network is obtained as the likelihood of this specific network divided by the sum of the likelihoods of all possible networks. Further statistics were calculated using these network probabilities. For example, the probability that individual A infected individual B was calculated as the sum of the probabilities of networks for which A infected B, while the probability that A infected a total of n other individuals was calculated as the sum of the probabilities of the networks in which A infected n other individuals.

Where in the initial subsetting of individuals disjoint subsets were identified, independent network reconstructions were calculated for each subset. Where the largest complete network in a subset omitted more than one individual from the subset, repeat calculations, removing those individuals inferred to be in that network, were performed, aiming to identify transmission events between other individuals in the subset.

## Computational considerations

Our algorithm runs at a combinatorial level of complexity dependent upon the number of individuals and the number of plausible paths through the network. We adopted different computational methods to identify maximum likelihood networks and statistical properties. For wards other than A, a rapid and comprehensive calculation of likelihoods for all plausible networks was possible. Uncertainty about properties of the network was then computed over the complete space of possible networks. For ward A, a first calculation was performed, generating likelihoods for a systematically chosen set of 0.2% of the plausible networks. From these networks, a heuristic likelihood threshold was chosen, identifying 10 networks with the highest likelihood values. These networks were used as starting points for independent downhill optimisation calculations. Given each network, the set of adjacent networks was identified, each constructed by the breaking and replacement of a single transmission event. Likelihoods were calculated for these networks, choosing the maximum likelihood network among these before repeating the calculation. Across these independent calculations, convergence to the reported maximum likelihood network was observed.

Uncertainty calculations for ward A were performed initially over the systematic sample of network space, plus networks sampled in the downhill optimisation, plus all networks one or two steps adjacent from the maximum likelihood network; this included a total of 14,040 networks. A second calculation of statistics describing network uncertainty was performed across all these networks but also including any remaining networks three steps adjacent from the maximum likelihood network; this included a total of 19,238 networks. Statistics derived from the smaller and larger ensembles of networks were then compared. The high degree of similarity between these statistics suggests convergence towards the true statistical average (*Figure 5—figure supplement 5*).

## Evaluating superspreading

Estimates of the distribution of the number of individuals infected by each person in each ward were used to assess the existence of superspreader individuals, comparing the fit of two models to the data.

Our network inference provided for each individual a distribution of the number of people infected, so we could say that each individual i infected j others with probability $p_{ij}$. We used these values to define 'latent data' to which we can fit Poisson models describing different hypotheses about heterogeneity in transmission rates. Under a model with no superspreading, the number of individuals infected by each person would be expected to follow a Poisson distribution with some rate R of transmission. The marginal likelihood of this model, defined as the Poisson likelihood integrated over the distribution of the latent number of individuals infected by each person, is derived as

$$\log L_1 = \sum_i \log \sum_j p_{ij} \left( \frac{r^j e^{-r}}{j!} \right)$$

This was maximised to estimate the common transmission rate r. We compared this hypothesis to an alternative, negative binomial model. This is equivalent to a model where the transmission rate is different for each person, so that each person infects a number of people described by a Poisson with a random rate drawn from a gamma distribution. We found parameters p and r maximising the marginal likelihood

$$\log L_2 = \sum_i \log \sum_j p_{ij} \frac{r+j-1}{r-1} (1-p)^j p^r$$

The marginal likelihoods $L_1$ and $L_2$ were then compared using the Bayesian Information Criterion (*Schwarz, 1978*) to account for the additional parameters in the second model.

## Source of patient infections

In the maximum likelihood networks, of 22 cases of patients being infected, two were inferred to originate from HCWs, with 20 of these being infected by other patients. We compared these values using a one-tailed test, calculating the cumulative density function of a binomial distribution with N=22 and probability 0.5 for the observed sample.

## Data availability statement

This study utilised 129 SARS-CoV-2 genomes from 98 individuals across the five outbreak wards. The COG-UK and GISAID sequence IDs for these samples are shown in *Supplementary files 1* and *2*. Genomic data are publicly accessible through the COG-UK website data section (https://www.cogconsortium.uk/data/) and GISAID (https://www.gisaid.org/). Sequences generated through the COG-UK consortium have associated public metadata, including age, sex, collection date (if available), and location to the level of UK county. COG-UK samples are sequenced under statutory powers granted to the UK Public Health Agencies. Matched patient data is securely released to the COG-UK consortium under a data sharing framework which strictly controls the handling of patient data. Information on whether individuals are healthcare workers or patients, and groupings of patients into their shared ward locations in hospital, are not for public release linked to their sequencing identifiers (eg. COG-UK sequence codes). This is because of the risk of deductive disclosure, potentially compromising study participant anonymity. However, code to fully reproduce the transmission network analysis using anonymised metadata and altered SARS-CoV-2 sequences is available via GitHub at https://github.com/cjri/a2bnetwork (*Illingworth, 2021*; copy archived at swh:1:rev:2c08d1a789b7f1a9ce758a86db27fc3d78b9d003) If a researcher requires access to restricted metadata (including healthcare worker status and patient ward locations) linked to the COG-UK sequence codes, then this will require a formal data sharing agreement with the COG-UK Consortium and Cambridge University Hospitals NHS Foundation Trust (CUH). Data will only be shared for public health and research purposes, not for commercial enterprise, and only to individuals working at reputable research and public health institutions for which data security can be assured. Should this be required researchers should contact the study corresponding authors in the first instance.

## Acknowledgements

This work was funded by COG-UK, which is supported by funding from the Medical Research Council (MRC) part of UK Research and Innovation (UKRI), the National Institute of Health Research (NIHR) and Genome Research Limited, operating as the Wellcome Sanger Institute; We also acknowledge the support from the Wellcome (Senior Clinical Fellowship to MPW (ref: 108070/Z/15/Z), Senior Research Fellowship to SB (ref: 215515/Z/19/Z), Senior Fellowship to IG (ref: 207498/Z/17/Z); Collaborative Grant to CJH (ref: 204870/Z/16/Z)); the Academy of Medical Sciences and the Health Foundation (Clinician Scientist Fellowship to MET), the National Institute for Health Research Cambridge Biomedical Research Centre at the Cambridge University Hospitals NHS Foundation Trust (to BW, MET); the NIHR Clinical Research Network Greenshoots award (to EGK); and MRC core funding (MC_UU_00002/11, for CJRI, DDA). CJRI acknowledges funding from Deutsche Forschungsgemeinschaft (DFG) Grant SFB 1310.

## Additional information

### Funding

| Funder | Grant reference number | Author |
| --- | --- | --- |
| Wellcome Trust | 108070/Z/15/Z | Michael P Weekes |
| Wellcome Trust | 215515/Z/19/Z | Stephen Baker |
| Wellcome Trust | 207498/Z/17/Z | Ian G Goodfellow |
| Wellcome Trust | 204870/Z/16/Z | Charlotte J Houldcroft |
| Academy of Medical Sciences | | M Estée Török |
| NIHR | | Ben Warne |

| | | M Estée Török |
|---|---|---|
| National Institute for Health Research | | Effrossyni Gkrania-Klotsas |
| Medical Research Council | MC_UU_00002/11 | Christopher J R Illingworth<br>Chris Jackson<br>Daniela de Angelis |
| Deutsche Forschungsge-meinschaft | SFB 1310 | Christopher J R Illingworth |
| The Health Foundation | | M Estée Török |

The funders had no role in study design, data collection and interpretation, or the decision to submit the work for publication.

## Author contributions

Christopher JR Illingworth, Conceptualization, Software, Formal analysis, Validation, Visualization, Methodology, Writing - original draft, Writing - review and editing; William L Hamilton, Conceptualization, Data curation, Formal analysis, Validation, Investigation, Methodology, Writing - original draft, Writing - review and editing; Ben Warne, Matthew Routledge, Ashley Popay, Conceptualization, Data curation, Formal analysis, Investigation, Writing - review and editing; Chris Jackson, Formal analysis, Methodology, Writing - review and editing; Tom Fieldman, Investigation, Writing - review and editing; Luke W Meredith, Charlotte J Houldcroft, Myra Hosmillo, Aminu S Jahun, Sarah L Caddy, Anna Yakovleva, Fahad A Khokhar, Theresa Feltwell, Malte L Pinckert, Iliana Georgana, Yasmin Chaudhry, Dominic Sparkes, Nick K Jones, Sushmita Sridhar, Sally Forrest, Tom Dymond, Kayleigh Grainger, Chris Workman, Data curation, Investigation, Writing - review and editing; Laura G Caller, Data curation, Funding acquisition, Writing - review and editing; Grant Hall, Lucy Rivett, Data curation, Investigation; Martin D Curran, Surendra Parmar, Nicholas M Brown, Resources, Data curation, Investigation, Writing - review and editing; Mark Ferris, Resources, Data curation, Investigation; Effrossyni Gkrania-Klotsas, Data curation, Funding acquisition, Investigation, Writing - review and editing; Michael P Weekes, Resources, Data curation, Funding acquisition, Investigation; Stephen Baker, Resources, Data curation, Funding acquisition, Investigation, Writing - review and editing; Sharon J Peacock, Supervision, Funding acquisition, Writing - review and editing; Ian G Goodfellow, Resources, Supervision, Funding acquisition, Project administration, Writing - review and editing; Theodore Gouliouris, Conceptualization, Supervision, Validation, Project administration, Writing - review and editing; Daniela de Angelis, Conceptualization, Supervision, Project administration, Writing - review and editing; M Estée Török, Conceptualization, Data curation, Formal analysis, Supervision, Funding acquisition, Investigation, Writing - original draft, Project administration, Writing - review and editing

## Author ORCIDs

Christopher JR Illingworth ![ORCID] https://orcid.org/0000-0002-0030-2784
William L Hamilton ![ORCID] http://orcid.org/0000-0002-3330-353X
Charlotte J Houldcroft ![ORCID] http://orcid.org/0000-0002-1833-5285
Myra Hosmillo ![ORCID] http://orcid.org/0000-0002-3514-7681
Aminu S Jahun ![ORCID] http://orcid.org/0000-0002-4585-1701
Sarah L Caddy ![ORCID] http://orcid.org/0000-0002-9790-7420
Grant Hall ![ORCID] http://orcid.org/0000-0003-3928-3979
Iliana Georgana ![ORCID] http://orcid.org/0000-0002-8976-1177
Lucy Rivett ![ORCID] http://orcid.org/0000-0002-2781-9345
Nick K Jones ![ORCID] http://orcid.org/0000-0003-4475-7761
Sushmita Sridhar ![ORCID] http://orcid.org/0000-0001-7453-7482
Mark Ferris ![ORCID] https://orcid.org/0000-0001-5040-4263
Effrossyni Gkrania-Klotsas ![ORCID] http://orcid.org/0000-0002-0930-8330
Nicholas M Brown ![ORCID] http://orcid.org/0000-0002-6657-300X
Michael P Weekes ![ORCID] http://orcid.org/0000-0003-3196-5545
Stephen Baker ![ORCID] http://orcid.org/0000-0003-1308-5755

Sharon J Peacock https://orcid.org/0000-0002-1718-2782
Ian G Goodfellow https://orcid.org/0000-0002-9483-510X
M Estée Török https://orcid.org/0000-0001-9098-8590

### Ethics

Human subjects: This study was conducted as part of surveillance for COVID-19 infections under the auspices of Section 251 of the NHS Act 2006. It therefore did not require individual patient consent or ethical approval. The COG-UK study protocol was approved by the Public Health England Research Ethics Governance Group (reference: R&D NR0195).

### Decision letter and Author response

Decision letter https://doi.org/10.7554/eLife.67308.sa1
Author response https://doi.org/10.7554/eLife.67308.sa2

## Additional files

### Supplementary files

• Supplementary file 1. GISAID sequence identifiers for the genomes used in this study.

• Supplementary file 2. COG-UK sequence identifiers for the genomes used in this study.

• Transparent reporting form

### Data availability

The SARS-CoV-2 genomic data used in this project are available via the COVID-19 Genomics Consortium UK https://www.cogconsortium.uk/data/ and the GISAID Initiative https://www.gisaid.org/ (Supplementary files 1 and 2). Source data files have been provided for Figures, including original data where figures are made from processed data, and where possible (some original files are very large). Symptom onset and location data for individual patients and health care workers are available from the authors on request; we are not able to share any data publicly which could lead to the identification of individuals in the study. The code used in this project is available from https://github.com/cjri/a2network (copy archived at https://archive.softwareheritage.org/swh:1:rev:2c08d1a789b7f1a9ce758a86db27fc3d78b9d003).

The following datasets were generated:

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
