## [Decision Letter]

**Acceptance summary:**

The reviewers agreed that your manuscript was an impressive piece of work, that provided a rigorous and granular perspective on infection dynamics that integrated data of various kinds: clinical, genomic, and demographic. Using these varied approaches, the study provides strong evidence for the importance of superspreading as a key feature of outbreaks. It is our hope that studies that similarly integrate data of various kinds might become the standard for resolving the particulars of outbreaks.

**Decision letter after peer review:**

Thank you for submitting your article "Superspreaders drive the largest outbreaks of hospital onset COVID-19 infections" for consideration by *eLife*. Your article has been reviewed by 2 peer reviewers, and the evaluation has been overseen by a Reviewing Editor and Aleksandra Walczak as the Senior Editor. The following individual involved in review of your submission has agreed to reveal their identity: Sarah E Cobey (Reviewer #2).

Summary:

This study of COVID-19 outbreaks in a major hospital makes two major contributions. First, it suggests that most hospital-acquired infections do not lead to further spread, and that superspreading events occur infrequently. Second, it advances analytic methods for reconstructing transmission networks by combining data on symptoms, behavior, testing, and viral sequences. This will facilitate analyses of other pathogen outbreaks.

Essential Revisions:

Clarifications on some aspects of the HCWs as outlined by the reviewers. Though their inclusion was a strength and focus on their paper, there are certain aspects that were not immediately clear, that readers could benefit from understanding.

These include, amongst others, questions about the conditions under which the HCWs and patients were tested over time, questions about the immune status of HCWs, and questions about the location function for HCW.

Pay special attention to the detailed questions raised by Reviewer #2. I consider these changes to be essential, and I shared some of those curiosities. Make no mistake, however, the reviewers and reviewing editor agree that this manuscript has very strong potential. It is a compelling study with an impressive mix of data sources and methods.

*Reviewer #1:*

Illingsworth et al. write a compelling and well motivated manuscript on the importance of superspreading in hospital settings.

It is a very well written and important piece of work with methodological advances beyond the application presented here.

I am not an expert in viral phylogenetics and will not be able to speak to the robustness of the evolutionary analysis performed.

This work is particularly strong in its adaptation of statistical techniques to identify transmission networks in highly sampled environments. Even though this is of great interest for some settings I wonder whether the authors had the ability to test their models predictive performance (i.e., if the authors withheld some data and tried to explain future case trajectories in the hospital). This may well be out of scope for this analysis but would make the results be generalisable beyond one particular setting. It would also provide additional support for the robustness of the method. It is very hard to evaluate whether the transmission networks reconstructed are indeed explained by the model applied here given the large number of parameters and somewhat long time of infectiousness of SARS-CoV-2.

The authors state that the sampling of infections within wards was nearly complete which is a major strength of this paper. Usually there are large biases in genomic sampling. It is remarkable that even asymptomatics here are sampled and sequenced if/when positive. However, there is asymmetry in sampling: within the wards it is high but what could the contribution of visitors have been?

Whereas the authors work is very illustrative I wondered after reading it whether there should be more information about the patients characteristics (age, sex, symptoms etc.).

Overall, I enjoyed reading this work!

*Reviewer #2:*

This study investigates the spread of SARS-CoV-2 in several wards of a hospital in the spring of 2020, aiming to measure transmissibility among patients and healthcare workers (HCWs). The investigators ambitiously incorporate seemingly all relevant data sources into a unified statistical framework in their attempt to reconstruct transmission networks.

The conclusions of the study appear well supported by the data, assuming a few uncertainties in my reading (discussed elsewhere) are resolved. A strength of the analysis is that transmission events are linked by not only the spatiotemporal proximity of individuals' infections but also the genetic similarity of their viruses, and this is a relatively circumscribed population (i.e., patient movement and HCW shift assignments are well known). This is thus an unusually well resolved transmission network over an extended period.

One of the study's claims appears weakly supported by the current analysis. The authors investigate whether Ct values, a measure of viral load, correlate with superspreading. Since Ct values are well known to vary over the course of an infection, some adjustment should probably be made for time of symptom onset, and asymptomatics potentially excluded. Otherwise there is a risk that any correlation between Ct values and increased transmissibility will be masked by variability in relative sampling times. (Transmissibility should probably not be assigned a binary variable either.) Relatedly, correlations between transmissibility and presence of, e.g., fever, would require a larger analysis, and I suggest the authors qualify their language about "identified risk factors."

An important public health consideration is whether it is appropriate to describe these as superspreading individuals or superspreading events. My impression is that the two are not clearly identifiable here, but the authors seem well positioned to address this ongoing debate directly. Additionally, it seems possible that the statistical breakdown estimated here (80% of cases are caused by 20% of infections) is distorted by the investigators' decision to analyze the wards with the largest outbreaks; if other wards were included, the estimated contribution of superspreading might be lower.

This paper presents a real methodological contribution in showing how diverse data sources can be integrated in a statistically coherent way. This will be invaluable in future outbreak investigations and will (I hope) motivate better surveillance, especially collection of date of symptom onset + sequences. In addition to an unusually clear description of the methods, especially for analyses of this complexity, the authors have commendably made their code publicly available, and it appears to be well documented. Future work will likely relax some of the assumptions here, e.g., by allowing unobserved intermediate infections and potentially allowing greater variation in mutation rates and serial intervals in immunocompromised hosts.

This work appears technically sound, but the following clarifications/checks would be really useful:

1. Under what conditions were HCWs and patients tested in the wards over time? How confident can we be that unobserved infections did not occur? Was there regular screening for asymptomatics (as suggested in the Discussion, lines 268-270)? When were visitors allowed?

2. Did the authors confirm that separate models were justified for the wards? Was there evidence of transmission b/w wards, perhaps involving the same HCWs?

3. In interpreting the transmissibility and susceptibility of HCWs and patients, is there any reason to think that many of the HCWs (especially in the red ward) might already have immunity from prior infection?

4. Are the results affected substantially if we assume that the location function for the HCWs is +/- 0.5 d instead of 1 d? The 1-d assumption could implicate HCWs unnecessarily. Fomite transmission seems hard to justify.

Further clarifications suggested for the text:

5. Please include symptom definitions and whether they varied by case type (HAI, HCW) and also describe the wards a bit more fully, if possible (square footage? patients per room?)

6. It would be really nice to have a time series of infections by ward relative to some arbitrary day 0. I think there should be some way to do this that would not violate privacy.

7. In the Discussion (lines 257-278), the authors recommend masks and hand hygiene. Are there any citations for the latter? If not, I might say something more vague (or mention only the former, with references).

8. I found the description of the network analysis of patient bed movements hard to follow (paragraph starting l. 373). What "network analysis" does FoodChain Lab do exactly? The listed criteria seemed straightforward to apply.

9. Line 431: Separate references and CIs here would help.

10. Line 445: Typo in the lognormal component → "((log(x) -mu)^2)"

11. Lines 658-661: Subscript typo in this criterion?

12. Lines 693, 696: There seems to be a major contradiction between "all possible networks" and all networks meeting the condition where individual i infects n others.

13. Is there any way to add schematics for the later steps of the analysis? I greatly appreciated what the authors already provided.

---

## [Author Response]

Essential Revisions:Clarifications on some aspects of the HCWs as outlined by the reviewers. Though their inclusion was a strength and focus on their paper, there are certain aspects that were not immediately clear, that readers could benefit from understanding.These include, amongst others, questions about the conditions under which the HCWs and patients were tested over time, questions about the immune status of HCWs, and questions about the location function for HCW.Pay special attention to the detailed questions raised by Reviewer #2. I consider these changes to be essential, and I shared some of those curiosities. Make no mistake, however, the reviewers and reviewing editor agree that this manuscript has very strong potential. It is a compelling study with an impressive mix of data sources and methods.

We thank the reviewers for their comments on our work. In addition to the responses to

reviewers noted below we note a slight change in the underlying A2B-COVID software

package, which has been in development in parallel to this project and is used for the

pairwise assessment of events. This change leads to our method being fractionally more

inclusive when assessing events in a pairwise manner, leading to more transmission events being inferred, but without a substantial change in the overall results.

Reviewer #2:This study investigates the spread of SARS-CoV-2 in several wards of a hospital in the spring of 2020, aiming to measure transmissibility among patients and healthcare workers (HCWs). The investigators ambitiously incorporate seemingly all relevant data sources into a unified statistical framework in their attempt to reconstruct transmission networks.The conclusions of the study appear well supported by the data, assuming a few uncertainties in my reading (discussed elsewhere) are resolved. A strength of the analysis is that transmission events are linked by not only the spatiotemporal proximity of individuals' infections but also the genetic similarity of their viruses, and this is a relatively circumscribed population (i.e., patient movement and HCW shift assignments are well known). This is thus an unusually well resolved transmission network over an extended period.One of the study's claims appears weakly supported by the current analysis. The authors investigate whether Ct values, a measure of viral load, correlate with superspreading. Since Ct values are well known to vary over the course of an infection, some adjustment should probably be made for time of symptom onset, and asymptomatics potentially excluded. Otherwise there is a risk that any correlation between Ct values and increased transmissibility will be masked by variability in relative sampling times. (Transmissibility should probably not be assigned a binary variable either.) Relatedly, correlations between transmissibility and presence of, e.g., fever, would require a larger analysis, and I suggest the authors qualify their language about "identified risk factors."

We have used more cautious language in this paragraph, now saying that the study “may suggest possible risk factors” for the specific cases examined, but that the dataset is too small to draw general conclusions on superspreader characteristics. We have also

emphasised this in the Discussion paragraph on study limitations:

The number of superspreader individuals identified in this study (five) is too small to draw general conclusions on superspreader characteristics. Moreover, it is not possible to disentangle whether superspreading was driven mainly by individual factors (such as infectivity or behaviour) or environmental factors (such as patient placement and ventilation at time of peak infectivity), or a combination of these. PCR cycle threshold (Ct) can vary for many reasons including the timing of sampling during COVID-19 infection, sampling type and technique, viral transport, sample preparation and variability between PCR runs. The finding that Ct values did not vary significantly between superspreader and non-superspreader individuals should therefore be interpreted with caution.

An important public health consideration is whether it is appropriate to describe these as superspreading individuals or superspreading events. My impression is that the two are not clearly identifiable here, but the authors seem well positioned to address this ongoing debate directly. Additionally, it seems possible that the statistical breakdown estimated here (80% of cases are caused by 20% of infections) is distorted by the investigators' decision to analyze the wards with the largest outbreaks; if other wards were included, the estimated contribution of superspreading might be lower.

Our model identifies individuals who appear to transmit the virus to a greater number of individuals than expected, and we identify a statistical pattern of overdispersion in the number of infections caused per individual. However, as above, we have included as a limitation in the Discussion the difficulty in disentangling between superspreader individuals vs superspreader events. With a relatively small set of superspreaders (five) our data cannot produce robust characteristics of superspreaders and we therefore do not reach firm conclusions on this.

This paper presents a real methodological contribution in showing how diverse data sources can be integrated in a statistically coherent way. This will be invaluable in future outbreak investigations and will (I hope) motivate better surveillance, especially collection of date of symptom onset + sequences. In addition to an unusually clear description of the methods, especially for analyses of this complexity, the authors have commendably made their code publicly available, and it appears to be well documented. Future work will likely relax some of the assumptions here, e.g., by allowing unobserved intermediate infections and potentially allowing greater variation in mutation rates and serial intervals in immunocompromised hosts.

We accept that our method is a starting point in integrating different forms of data and hope to push back some of its current limitations in future work.

This work appears technically sound, but the following clarifications/checks would be really useful:1. Under what conditions were HCWs and patients tested in the wards over time? How confident can we be that unobserved infections did not occur? Was there regular screening for asymptomatics (as suggested in the Discussion, lines 268-270)? When were visitors allowed?

Patients and HCW had separate testing criteria and sample workflows. HCW were tested in the CUH HCW screening programme, which included both asymptomatic screening and symptomatic testing arms. Asymptomatic screening at the time of this study was focused on staff working on COVID-19 "red" wards (designated for patients with confirmed COVID-19 infection), wards with hospital-acquired infection outbreaks, and wards with high rates of staff positivity. Suggested symptomatology to prompt staff testing are described in Rivett et al., divided into "major" criteria (eg. fever and/or new persistent cough) and "minor" criteria (eg. coryzal symptoms, headache, myalgia) (see Table 1 of Rivett et al., eLife 2020;9:e58728).

For all five of the outbreaks described in this study, all staff working on the outbreak wards were screened by the CUH HCW screening team (i.e. tested regardless of any symptoms or if asymptomatic) during the outbreak periods (prompted by the outbreak investigations and/or high rates of staff positivity on the wards). It is theoretically possible that HCW could have caught COVID-19 early on in the outbreaks and cleared the virus quickly, becoming negative at time of testing, or caught the virus asymptomatically after the screening test, or had levels of virus below the detection limit of the assay (and thus have been false negatives). However, levels of SARS-CoV-2 RNA below the assay detection limit are unlikely to be infectious (and thus not significant for the inferred transmission networks).

There was not systematic asymptomatic screening for inpatients in CUH during the study period, but targeted patient screening on wards with hospital-onset COVID-19 outbreaks was performed. Ward E was a COVID-19 “red” ward; all patients on this ward had tested positive prior to placement there. Wards A-D were all “green” wards (designated for non-COVID-19 patients) at the time the hospital-onset COVID-19 outbreaks started. For Ward A, symptomatic contacts of confirmed cases were tested initially, and as the outbreak grew and more cases were confirmed, ultimately all patients on the ward were screened (including asymptomatics). For Ward C, contacts of confirmed positives and/or patients who developed symptoms were screened, and for Ward D, symptomatic contacts of the index case were tested. Thus, for Wards C and D, systematic asymptomatic screening of all patients on the ward during the outbreaks was not performed, and it is possible some asymptomatic infections (that could have contributed to the transmission networks) were missed. We have added this as a study limitation in the discussion. For the Ward B outbreak, when the index patient (case B0) tested positive, all staff members who had worked on Ward B within the preceding 2 weeks plus all patients on that ward were screened (regardless of any symptoms). Thus, there is high confidence for Wards A, B and E that all infections among both staff and patients were detected (providing the amount of viral RNA was sufficient for detection).

Visitor restrictions were introduced 25th March 2020 and so were present for almost all of this study (first positive swab was for Ward E, collected a few days before this). Visitors were only permitted in exceptional circumstances: for patients at the end of life, visitors with a direct care role for the patient, or the parent of a child (not relevant here, as all of the wards were for adult patients).

The above has been added as a new section to the Methods, section ‘Testing criteria’. The issue of missed infections in the transmission networks has been expanded on in the Discussion, paragraph starting ‘We acknowledge several limitations to our study’.

2. Did the authors confirm that separate models were justified for the wards? Was there evidence of transmission b/w wards, perhaps involving the same HCWs?

Separate models were justified by the data we have, as shown in a new Figure 1. While no HCWs worked on multiple outbreak wards, we cannot exclude unrecorded staff contact outside of these wards (during lunch, outside of work, etc.) We now preface our analysis with a pairwise analysis of the data across all wards. This analysis suggests that

transmission between individuals on different wards was unlikely, but cannot definitively rule out such an event having occurred.

3. In interpreting the transmissibility and susceptibility of HCWs and patients, is there any reason to think that many of the HCWs (especially in the red ward) might already have immunity from prior infection?

A study of SARS-CoV-2 serology conducted at Cambridge University Hospitals NHS

Foundation Trust found that, between 10th June and 7th August 2020, 410/5,698 (7.2%) staff tested positive for SARS-CoV-2 antibodies

(https://papers.ssrn.com/sol3/papers.cfm?abstract_id=3724855). As expected, prevalence was higher for staff working in COVID-19 "red" wards than in “green” wards (9.47% versus 6.16%). Not all people with detectable SARS-CoV-2 antibodies will have clinically protective immunity. Our study of the five “outbreak” wards was based earlier in the pandemic than this serology study (March – June 2020), when staff seroprevalence would have been lower. The proportion of staff with neutralising antibodies during the outbreaks studied here was therefore low, likely <5% in March 2020. Thus, it likely played a minor role in transmission dynamics. This has been added to the discussion, paragraph starting ‘Our inferences of transmission were performed on the basis of a largely complete dataset’.

4. Are the results affected substantially if we assume that the location function for the HCWs is +/- 0.5 d instead of 1 d? The 1-d assumption could implicate HCWs unnecessarily. Fomite transmission seems hard to justify.

This does not make a large difference to the results obtained. In the framework of this

method we consider time in discrete intervals of one day. Under the default setting our

method assigns HCWs as being present on a ward with probability one half during an

interval of +/- 1 days from any days on which they are recorded as being present. This was designed to account for fomite transmission, but also for multiple other factors, including the possibility of night shifts spanning more than one day, but being recorded as presence on a single day. We have now generalised our method to allow this probability to be input as an arbitrary value.

We reran calculations in which this probability was set to zero. This leads to some changes to the details of the inference, as shown in a new Figure 5S1, but no change in the headline result of 21% of infections causing 80% of transmission events.

Further clarifications suggested for the text:5. Please include symptom definitions and whether they varied by case type (HAI, HCW) and also describe the wards a bit more fully, if possible (square footage? patients per room?).

Re: Symptom definitions – these were collected separately for HCW and patients. HCW

testing was part of the CUH HCW screening programme, and testing criteria are described in Rivett et al., (Table 1). Staff who tested positive were then contacted by members of the HCW screening team and asked retrospectively about their symptoms, and onset dates were recorded by the HCW screening team and used in this analysis. For patients, symptom onset dates were collected by retrospective review of patients’ electronic hospital records (EPIC Systems), usually noted at presentation by the clerking doctor but all records were examined for any suggestion of symptoms. Symptom definitions followed the standard national recommendations at the time: initially fever, breathlessness and new continuous cough; muscle aches was added in early May and loss of taste or smell was included from mid-May.

This has been added to Methods, ‘Data collection’ section.

Re: Ward descriptions – We are unable to provide precise information about the wards in order to preserve confidentiality. The wards are typically organised into bays, with 4 to 6 patients per bay, and a limited number of side rooms (which are critical for infection control purposes). In summary, Ward A had 30 beds with 3 side rooms; Ward C had 27 beds with 3 side rooms; Ward D had 30 beds with 4 side rooms; Ward E had 26 beds with 3 side rooms. The Ward B outbreak was focused around a ward with 32 beds of which 12 were side rooms, although several adjacent wards were screened as part of the outbreak investigation (as staff were shared between these wards).

This has been added to Methods, ‘Sample selection’ section.

6. It would be really nice to have a time series of infections by ward relative to some arbitrary day 0. I think there should be some way to do this that would not violate privacy.

We have added a new Figure 5 containing symptom onset dates and dates at which

sequence data were collected, noting that not all samples led to high quality sequences. This information has been normalised to day 0 for each ward; this is the maximum we are able to make public given the other information we share.

7. In the Discussion (lines 257-278), the authors recommend masks and hand hygiene. Are there any citations for the latter? If not, I might say something more vague (or mention only the former, with references).

This is in accordance with PHE guidance. The specific question of the role of hand hygiene has been addressed by the UK Government Scientific Advisory Group for Emergencies (SAGE) here:

https://assets.publishing.service.gov.uk/government/uploads/system/uploads/attachment_data/file/897598/S0574_NERVTAG-EMG_paper_-_hand_hygiene_010720_Redacted.pdf

However we have made the sentence more generally refer to infection control practices:

First, scrupulous adherence to infection control practices including appropriate

personal protective equipment (PPE) at all times, even on green wards and in nonclinical hospital areas, is required to limit transmission between asymptomatic patients and staff in which COVID-19 is not suspected.

8. I found the description of the network analysis of patient bed movements hard to follow (paragraph starting l. 373). What "network analysis" does FoodChain Lab do exactly? The listed criteria seemed straightforward to apply.

SQL was used to identify patient co-locations that met our set parameters and these were then imported into FoodChain Lab to draw the network diagram.

We have expanded on this paragraph to make the methods clearer, new text in red font (in Methods, section ‘Sample selection’):

An initial network analysis of patient bed movements was conducted to add patients to each ward cluster that may have been in contact with the HAI cases, either with community onset infections on the same ward or while they were co-located on other wards outside of the “outbreak wards” themselves. This analysis was undertaken in two steps using SQL v18.5.1 and FoodChain-Lab (an extension of the Knime Analytics Platform v3.6.1). The first step involved utilising SQL to process case and ward movements data from CUH patients, creating a list of ward-based case co-locations that were within the set parameters of the network analysis model. These parameters were (1) an infectious period that included the 4 days prior to symptom onset up until 7 days after symptom onset, and (2) a susceptibility period of 14 days prior to symptom onset. Where symptom onset date was not available, the case positive specimen date was used instead. The second step was to import the case colocations data into FoodChain-Lab, which was used to draw a social network diagram of cases and their ward co-locations with one another that met the set parameters. This network diagram was used to identify any clustering of cases by ward that had not met the original criteria of HAI and HCW cases but could have been involved in shared transmission with those individuals based on their co-location within the infection period of the virus. This yielded the final case set for each ward cluster taken forward for analysis using the transmission reconstruction model.

9. Line 431: Separate references and CIs here would help.

We have separated the references in order to clarify the origin of these parameters. We

could not find confidence intervals for these parameters in the literature, albeit they could be obtained by repeating the work of the papers cited. For the purpose of our study we utilised the estimated parameters from the literature cited.

10. Line 445: Typo in the lognormal component → "((log(x) -mu)^2)"

Fixed.

11. Lines 658-661: Subscript typo in this criterion?

Fixed.

12. Lines 693, 696: There seems to be a major contradiction between "all possible networks" and all networks meeting the condition where individual i infects n others.

We don’t think there is a contradiction and have rewritten these lines to clarify the matter. Given likelihoods for all possible networks, the probability of a specific network was calculated as its likelihood divided by the sum of all likelihoods. The probability that individual i infects n others is then the sum of the probabilities of all of the networks in which i infects n others.

13. Is there any way to add schematics for the later steps of the analysis? I greatly appreciated what the authors already provided.

We have added another figure (5S4) showing how sequence variants constrain which

transmission events are possible, and the order in which they occur; we think this provides clarification of two points in the methods.